# Influence of Gender, Parental Control, Academic Performance and Physical Activity Level on the Characteristics of Video Game Use and Associated Psychosocial Problems in Adolescents

**DOI:** 10.3390/bs14121204

**Published:** 2024-12-15

**Authors:** Manuel Isorna-Folgar, Adrián Mateo-Orcajada, José María Failde-Garrido, María Dolores Dapia-Conde, Raquel Vaquero-Cristóbal

**Affiliations:** 1Departamento de Análisis e Intervención Psicosocioeducativa, Facultad de Educación y Trabajo Social, Universidad de Vigo, 32004 Ourense, Spain; isorna.catoira@uvigo.es (M.I.-F.); jfailde@uvigo.es (J.M.F.-G.); ddapia@uvigo.es (M.D.D.-C.); 2Facultad de Deporte, UCAM Universidad Católica de Murcia, 30107 Murcia, Spain; 3Research Group Movement Sciences and Sport (MS&SPORT), Department of Physical Activity and Sport, Faculty of Sport Sciences, University of Murcia, 30100 Murcia, Spain; raquel.vaquero@um.es

**Keywords:** academic performance, adolescents, gender, parents, physical activity level, psychology, video games, youth

## Abstract

Previous research has determined the relevance of video games for adolescents; however, it has not been possible to establish differences in usage patterns and certain psychological variables according to gender, parental control, academic performance, physical activity level or game type, nor the relationship between these variables. For this reason, the aims of this research were as follows: (a) to determine the differences in the gaming variables and the psychological variables related to video games according to gender, the closest environment, the academic performance and the level of physical activity; and (b) to determine which gaming variables and behavioral variables influence psychological variables in adolescents. A descriptive, cross-sectional study was carried out involving 2567 adolescents (mean age: 15.06 ± 2.81 years). Participants completed eight questionnaires on the study variables. The results showed that males play more video games than females; play different types of games and on different platforms; and have more psychological problems than females. Having separated parents and having a greater parental control over video game use is associated with more time spent playing video games. Poorer academic performance is related to playing shooters and open-world games, as well as with a more negative emotional response. However, playing shooters and sport and racing games is related to more physical activity. Playing online games, mainly with strangers, is related to higher addictive and problematic uses. In addition, the times of use during the week and on weekends, especially on weekends, stand out as predictors of most psychological variables related to video games in adolescents. This study provides further scientific evidence on the role of certain behavioral and game-related variables in the relationship between video games and well-being. In addition, it highlights the importance of analyzing in the future those video game players who do not have a problematic or addictive use of video games, but who play frequently without any associated problem. From a practical perspective, the promotion of video games with social and cooperative components, or those that promote physical activity, could be related to social and psychological benefits.

## 1. Introduction

Video games have established themselves as one of the main forms of modern leisure, especially for the adolescent population aged between 12 and 17, with between 70 and 75% playing some form of video game on different platforms, reflecting the deep integration of video games into the daily lives of this population [1]. Previous studies have showed that excessive and addictive use can have negative consequences on the mental health and well-being of adolescents [2]. Furthermore, more time spent playing video games could decrease the time spent studying, which could affect the academic performance of adolescents who spend more time playing games [3]. Also, more time spent playing video games could decrease the time available for other activities, such as physical activity, increasing the risk of physical inactivity, sedentary lifestyles and overweight, obesity or other metabolic disorders [4]. However, in both cases, there are studies with contradictory results [5,6], which calls for further research in this area. In addition, it should be noted that patterns of play may depend on different socio-demographic variables [7,8]; but knowledge about the interaction of these variables is still lacking.

### 1.1. Psychological Theories Explaining the Impact of Video Games on Adolescents

It is important to be aware that the psychological attraction generated by video games is largely due to the ability to satisfy the basic psychological needs for competence, autonomy and social relatedness [9]. This is based on self-determination theory (SDT), which provides a useful theoretical framework for analyzing how satisfying the basic psychological needs increases the intrinsic motivation for a task [10]. As a consequence, there is an increase in the priority the person gives to that activity and the time he/she devotes to it, a phenomenon that has been widely demonstrated to occur during the use of video games [11].

Therefore, in order to understand the reasons why adolescents are attracted to video games, it is necessary to consider the motivations underlying gaming behavior. This approach explains how video games can provide an environment where players experience a sense of achievement, control over their decisions and social relatedness, satisfying these basic needs. This helps to understand why video games not only motivate players to continue playing but can also lead to an improvement in psychological well-being by satisfying these needs [12]. In contrast, a mismatch in this motivation can lead to problematic patterns of use [13,14].

On the other hand, it is worth noting that, according to the uses and gratification theory, players’ motivation is key because it influences their behavioral intentions and subsequent choices [15]. Based on this, this theory points out that users structure their behavior in response to their specific needs to obtain gratification [16]. Previous research supported by this theory in the field of video games has shown that personal and social integrative benefits are the best predictors for engagement and purchase [17]. Similarly, enjoyment and achievement are the factors that influence the intention to continue playing, so that hedonic and utilitarian gratifications are the ones that most affect this intention [18,19].

### 1.2. Impact of Video Games on Academic Performance and Its Relationship with Psychological Variables in Adolescents

In the academic field, it has been observed that addictive video game use decreases academic performance [3], but may also produce benefits in cognition and learning in adolescents [6]. Moreover, previous research has shown that playing video games alone has a minimal influence on academic performance, with other factors such as cyberbullying or exposure to violent content being the most relevant [20]. The timing of play seems to be relevant in this respect, as playing before school seems to produce negative effects on academic performance, but not playing after school [21]. Therefore, there is a lack of studies on the time spent, type and/or form of play that might affect the alterations produced in academic performance [22].

In the psychological field, the importance of academic performance lies in the fact that it is fundamental to adolescent health, as it is related to health-risk behaviors [23], mental health [24] and even sleep quality [25]. However, no previous research has explored the relationship of other psychological variables that could be affected using video games with academic performance, and this is important given the interdependence between these three factors and their influence on future health.

### 1.3. Impact of Video Games on Physical Activity Level and Its Relationship with Psychological Variables on Adolescents

As for the level of physical activity, previous research has shown a positive association between video game consumption and unhealthy lifestyle habits, such as a decrease in physical activity [26]. It is true that this association occurs mainly with non-active video games, as they keep the player seated for several hours, such as first-person shooter video games [5]. However, it has been observed that ball sport simulators (e.g., FIFA or NBA), even though they are non-active video games, are associated with increased physical activity and participation in sports clubs [5]. Therefore, the results are not consistent across studies, because, although the use of video games is related to higher body mass and lower self-reported general health status, there is not enough evidence to associate gaming time with a worse level of physical activity [27].

In the psychological domain, adolescents with a higher level of physical activity show a better psychological state [28], but the impact of video game use on the psychological health of adolescents by level of physical activity is unclear. In this respect, the use of active video games seems to increase self-esteem and mental health in general, as well as social well-being, but it seems that the type of video game played is very influential [29,30]. On the other hand, sedentary video game use is associated with poorer well-being, isolation and aggressive behavior [31]. Therefore, playing video games and not being physically active seems to be detrimental to psychological health [32], but it is unknown what happens in adolescents who play video games and are also physically active, as well as the psychological variables related to video games that are affected. For this reason, it is important to know which gaming-related variables and which psychological variables in relation to video games, beyond problematic or addictive use, are related to the level of physical activity of the players.

### 1.4. Socio-Demographic Factors That Determine the Use of Video Games and Their Impact on Psychological Variables in Adolescents

Although motivations and intention to play are important, during adolescence, differences are observed in gaming tendencies (time, type, form of play) and in the psychological impact of video game use as a function of socio-demographic variables such as gender [33] and the influence of the closest environment [34].

#### 1.4.1. Gender as a Determinant of the Use of Video Games and Its Impact on Psychological Variables in Adolescents

According to the gaming variables, it has been observed that the motivations and patterns of play between males and females are different, with males tending to play competitive video games, while females prefer social, non-competitive and simulation video games [7]. In addition, males tend to play for longer periods of time than females, perhaps because they report more fun and motivation [35]. Therefore, differences exist in terms of frequency and type of gaming, although both genders are susceptible to problematic gaming disorders [36].

In the psychological domain, differences are observed between male and female adolescents, with females tending to show greater psychological distress, anxiety and depression compared to males [37]. Furthermore, differences are observed in coping strategies, as well as in the effect of anxiety and stress on well-being and self-esteem in males and females [38]. The higher use of video games leads to males exhibiting more problematic use of video games compared to females (5.8% vs. 3.0%), thus classifying them as more addicted [39]. However, there are no major gender differences in this respect, and in both male and female adolescents, problematic video game use leads to negative psychological consequences such as increased aggression, depression or isolation [40]. Therefore, although gender differences in problematic use are observed, both males and females are affected, but it is not known whether other psychological variables related to video game use differ according to gender.

#### 1.4.2. Closest Environment as a Determinant of the Use of Video Games and Their Impact on Psychological Variables in Adolescents

The closest environment is another aspect of great relevance in the use of video games by adolescents [34]. Regarding the gaming variables, previous studies have shown that adolescents with balanced parental control and a stable family environment develop healthier gaming habits, while those with less supervision or from dysfunctional families are more likely to develop behaviors related to excessive video game use [41]. It is also worth noting that adolescents who play with siblings or friends mitigate the likelihood of excessive play time [8]. However, there are few studies that analyze the relationship between time of use, type of video game or platform used in relation to the closest environment.

According to the psychological domain, it has been observed that high levels of parental involvement, characterized by support and communication, are associated with better well-being, higher self-esteem and lower anxiety [42]; however, dysfunctional families can lead to increased psychological distress [42]. On the video games field, differences are also found in the psychological sphere, highlighting the fact that playing social games, in which one interacts with others, has more positive effects on well-being and is associated with less problematic game use compared to non-social games played individually [43,44,45]. Similarly, online versus offline play report differences, with online players showing more addictive behaviors, maladaptive cognitions, interpersonal conflicts and social isolation [46,47]. In addition, the people with whom adolescents play video games are also relevant, as playing cooperatively with friends shows more positive and prosocial behavior during play, increasing friendship quality [48], while playing with parents decreases internalizing and aggressive behaviors [49]. The fact that the social environment and the form of play influence psychological variables, such as prosocial behaviors and emotional satisfaction [50], increases the need to look more closely at aspects that may be related to the psychological state of adolescents.

### 1.5. Research Gap, Research Questions and Aims

Based on the previous scientific literature on the use of video games, it can be observed that the variables of the game itself (time, type of game, type of platform used) differ according to aspects such as gender and closest environment, and these variables may in turn be related to other aspects such as academic performance and level of physical activity [19,24,35,42]. More specifically, previous research has shown that problematic and/or addictive use of video games differs according to gender and closest environment; and this in turn may influence the academic performance and level of physical activity, with the consequences this has for the mental health of adolescents [23,29,38,47]. However, most adolescents do not engage in problematic or addictive video game use, as they play video games in a normal and appropriate way. Furthermore, information on this subject is scarce, with small samples of participants, and it is not possible to draw a reliable conclusion about the relationship between these variables. Therefore, more research is needed to analyze differences in gaming habits as a function of gender, closest environment, physical activity and academic performance with a large sample of adolescents to analyze the interaction between these factors.

On the other hand, psychological variables could also be affected by the use of video games and the variables of the game itself (time, type of game, type of platform used) [31,51]. However, the previous literature has not focused on these variables as influencing the psychological state, focusing on other factors such as gender, immediate environment, academic performance or physical activity [23,28,37,42]. Although there are studies that have analyzed how each of these variables separately affect the psychological state of adolescents, there are no known studies that have analyzed the interaction between these factors and the gaming variables, or the specific weight that these factors could have on mental health of this population. The relevance of this calls for research that focuses on the existing differences in psychological variables related to video games, considering socio-demographic variables such as gender or closest environment, and other behavioral parameters, such as academic performance and level of physical activity.

Based on the previous scientific literature, the research questions posed for this manuscript are (a) ‘What factors related to video game use (playing time, game type, platform) show differences according to gender, closest environment, academic performance and physical activity level in adolescents?’ (b) ‘How are psychological variables related with video game use (playing time, game type, platform), gender, closest environment, academic performance and physical activity level in adolescents?’ and (c) What are the factors that determine psychological variables in adolescents: gaming habits, physical activity or academic performance?’ The following aims are proposed to provide answers: (a) to determine the differences in the gaming variables and the psychological variables related to video games according to gender, the closest environment, the academic performance and the level of physical activity; and (b) to determine which gaming variables and behavioral variables influence psychological variables in adolescents.

## 2. Materials and Methods

### 2.1. Design

The present study was descriptive and cross-sectional and was conducted to measure the level of problematic video game use, addiction, motives for video game use, mood repair, passion scale and emotional and behavioral symptoms of playing video games in the adolescent population. The study design and protocol followed the STROBE statement [52]. All participants and their legal guardians signed the informed consent before the start of the research. They were informed of the procedure to be followed, as well as the questionnaires to be completed, and the aim pursued with the research. They were also informed that the treatment of the data would be completely anonymous and that only the project researchers would use the data. Participation was voluntary, and the participants were informed that they could withdraw from the research at any time, which would result in the elimination of their data collected up to that point. The ethics committee of the University of Vigo approved the research design (code: CE-DCEC-UVIGO-2022-10-04-5109) following the guidelines of the Declaration of Helsinki and the World Medical Association code.

### 2.2. Participants

The sample size was calculated using the EpiCalc package included in the Rstudio 3.15.0 statistical software (Rstudio Inc., Boston, MA, USA), following the methodology of previous research [53,54]. The standard deviation (SD) of previous research that analyzed the motives for playing video games in the adolescent population was used (SD = 1.07) [55]. Thus, for an estimated error (d) of 0.05, for a 95% confidence interval, the minimum sample needed to carry out the research was 1794 adolescents.

According to data from the Department of Education of the Xunta de Galicia, there are a total of 89,000 students enrolled in Compulsory Secondary Education, Baccalaureate or Vocational Training. From this population, a sample was selected by means of two-stage sampling, by conglomerates, for the selection of the first-level units. In this way, a total of 24 centers were randomly selected, both public and private/subsidized, urban and rural, and belonging to the four Galician provinces, considering the existing quotas at the population level.

All the participants met the following inclusion criteria: (a) age between 12 and 18 years old; (b) attending compulsory secondary education, baccalaureate and vocational studies; (c) volunteering to participate in the research; and (d) not presenting any incapacitating disease that prevented the completion of the questionnaires. The exclusion criteria were (a) not completing all the questionnaires in their entirety.

A total of 2567 adolescents met the inclusion criteria for this research. After applying the exclusion criteria, a total of 2513 adolescents (mean age: 15.06 ± 2.81 years old) ultimately participated in the research. The socio-demographic characteristics of the sample can be found in Table 1.

### 2.3. Instruments

#### 2.3.1. Socio-Demographic, Play Habits and Academic Performance Variables

A socio-demographic ad hoc questionnaire was used, in which participants were asked about their age, gender, academic performance, parental control of time spent playing video games, cohabitation of parents, type of video games played, platform most used to play video games, time spent playing during weekdays and weekends, usual way of playing and money spent per month on video games. This questionnaire was based on the one used in previous research [56].

#### 2.3.2. Physical Activity

A question was used to determine whether adolescents participated in leisure time physical sports activities or not. Those who indicated participation in physical sporting activities answered five additional questions that allowed for the calculation of the index or pattern of the amount of habitual physical sporting activity (Finnish physical sporting activity index) [57,58], which referred to frequency, duration, intensity and participation in organized sports and sports competitions. The Spanish version was used, which was previously validated in previous research, and had an adequate validity and reliability (Cronbach’s alpha = 0.81) [35,36]. The responses were recoded into three categories so that all of them had a similar weight for calculating the index or pattern. The resulting value ranged from 5 to 15. Lower scores are characteristic of less active individuals, while higher scores are indicative of more active individuals. In accordance with previous research conducted with adolescents using this same measurement [59,60], and to better represent the patterns of physical activity, the score was used to classify participants into vigorous, moderate, light and insufficient activity.

#### 2.3.3. Experiences Related to Video Games

The questionnaire of experiences related to video games (CERV) was used [37]. This questionnaire was designed in Spanish, and is composed of 17 items that allow the detection of problematic use of video games in adolescents. It is completed with a Likert scale ranging from 1 to 4 points (1: never/almost never; 4: almost always). This questionnaire has two dimensions (dependence/evasion and negative consequences). The dependence/evasion range is between 8 and 32 points, while negative consequences range between 9 and 36 points. Both dimensions showed a Cronbach’s alpha above 0.86, making this a valid and reliable instrument for use in adolescents. The total score is obtained by adding the scores of the items belonging to the two subscales. A higher score in the dimensions of the questionnaire, or in the final score, indicates a worse experience of video game use [61].

#### 2.3.4. Video Game Addiction

The video game addiction scale for adolescents (GASA) was used [38,39]. This instrument consists of seven items corresponding to a structure of 7 dimensions (salience, tolerance, emotion, relapse, abstinence, conflict and problems) that are grouped into a higher order factor: addiction. It is completed with a 5-point Likert scale (1: never; 5: very often). The final score ranges between 7 and 35 points. A higher score on this questionnaire indicates greater addiction. The reliability of this instrument is high, with a Cronbach’s alpha of 0.86. This instrument was designed and validated by Lemmens et al. [62] and was translated and validated in Spanish by Lloret et al. [63].

#### 2.3.5. Motives for Playing Video Games

The motives for playing video games were analyzed using the video gaming motives questionnaire (VMQ), which was initially designed in Spanish [40]. This questionnaire allows for the analysis of eight motives for playing video games: recreation, competition, cognitive development, coping, social interaction, violent reward, customization and fantasy. It is composed of 24 items, 3 per motive analyzed. It is answered with a 5-point Likert scale (0: do not agree at all; 4: strongly agree). The final score for each motive ranges from 0 to 12 points. A higher score indicates that the motive for gaming is greater. This instrument is considered valid and reliable to determine the motives for playing video games as the Cronbach’s alpha for each of the motives ranges between 0.76 and 0.93 [64].

#### 2.3.6. Emotional States

The Trait Meta-Mood Scale (TMMS-24) was used to assess emotional states. The 24-item version was adapted and validated in Spanish [65]. The scale contains three key dimensions: attention to feelings, emotional clarity and repair and regulation of mood, and it is evaluated with a five-point Likert-type scale (1 = strongly disagree, 5 = strongly agree). For this research, the repair dimension that refers to the subject’s belief about his/her capacity to interrupt and regulate negative moods and prolong positive ones was measured. This subscale consists of 8 items, with the final score ranging from 8 to 40 points. A higher score indicates a greater ability to interrupt and regulate negative moods. Fernández-Berrocal et al. [65] obtained Cronbach alphas of 0.86 for this dimension of the scale.

#### 2.3.7. Passion Towards the Activity

The passion scale [66] was used to measure passion towards the activity, using the validated Spanish version for this purpose [67]. This scale is divided into two subscales (harmonious and obsessive) composed of six items each. Thus, the scale is composed of a total of twelve items that are rated using a 7-point Likert scale (1: total disagreement; 7: total agreement). Both subscales ranged from 6 to 42 points. A higher score refers to a greater passion demonstrated for the activity. For the Spanish version, the validity and reliability of the scale as a whole, as well as of the subscales, demonstrated an adequate validity and internal consistency (0.81) [67].

#### 2.3.8. Social, Emotional and Behavioral Problems

The strengths and difficulties questionnaire (SDQ) [68] was used in its Spanish version [45]. It is composed of 25 items distributed into five subscales (5 items per subscale): emotional symptoms, conduct problems, hyperactivity, peer problems and prosocial behavior. A 3-point Likert scale (0: not true; 2: certainly true) was used for its completion. The score for each subscale ranges from 0 to 10 points. Higher scores show greater social, emotional and behavioral problems. The Spanish version has been validated, and it has a high reliability (0.88) [69].

### 2.4. Procedure

The data were collected in the classrooms of the respective education centers, with prior authorization from the management teams in groups of 15 to 25 individuals. For the completion of the questionnaires, the adolescents remained in a room isolated from the noise of the educational center, avoiding any distractions that could alter the answers. A questionnaire was administered, which each participant had to complete individually. The data collection was overseen by researchers, who had experience in conducting this type of task. The researchers present during the completion of the questionnaires did not condition the participants’ responses. The researchers simply limited themselves to resolving any doubts that might arise during this process, as well as reminding the participants that there were no good or bad answers to the questionnaires. The order in which the questionnaires were completed was randomized for each participant. The time required to complete the questionnaire ranged from 30 to 45 min.

### 2.5. Data Analysis

The Kolmogorov–Smirnov test was used to analyze the normality of the data, as well as skewness and kurtosis. Because the variables included followed a normal distribution, parametric tests were used for their analysis. First, the chi-square test (χ2) was performed to find the differences in the gaming variables as a function of the gender, closest environment, academic performance and physical activity level. Percentages and frequencies were used for the analysis. The corrected standardized residuals were used to determine significance, establishing ±1.96 as the reference value. Cramer’s V was used for the post hoc comparison of the 2 × 2 tables, and the Contingency Coefficient was used in the 2 × *n* tables, to obtain the statistical value. The maximum expected value was 0.707; r < 0.3 indicated a low association; r < 0.5 indicated a moderate association; and r > 0.5 indicated a high association [70]. Two Student’s *t*-tests were performed to analyze the differences in the psychological variables according to gender (male-female), and the cohabitation of the parents (together-not together). Cohen’s d was calculated to determine the effect size in these cases, being small when d < 0.2; moderate when d < 0.8; and large when d > 0.8 [71]. Four ANOVAs were performed to find differences in the psychological variables as a function of parental control of time use, academic performance, level of physical activity and form of play. The Bonferroni post hoc test was used to determine between which groups the differences were significant. Partial eta squared (ηp2) was used to calculate effect size, being small: ES ≥0.10; moderate: ES ≥ 0.30; large: ES ≥ 1.2; or very large: ES ≥ 2.0 [72]. Pearson’s correlation coefficient was used to determine the relationship between psychological variables, game-related variables and behavior-related variables. For the variables that were statistically significant in the Pearson correlation, a linear regression analysis was performed to predict the variables that could most influence the psychological state of the players. Statistical significance was established for a value of *p* < 0.05. The statistical analyses were performed with the SPSS v.25.0 statistical package (SPSS Inc. IL).

## 3. Results

### 3.1. Differences in the Gaming Variables According to Gender, Closest Environment, Academic Performance and Physical Activity Level

Table 2 shows the differences in the type of video games used, the platform of habitual use, the time spent during the week and during the weekend, the way of playing and the money spent per month on video games between males and females. The results show differences in the type of video game played (*p* < 0.001–0.017) and the platform used (*p* < 0.001). Regarding playing habits, males spent more time playing (*p* < 0.001), preferred to play online with friends (*p* < 0.001) and spent more money playing (*p* < 0.001) than females.

Differences according to whether the parents live together or separately are found in Table 3. Differences were found on the platform used (*p* = 0.020–0.001). The differences were significant in the time spent during the week, with the adolescents whose parents lived together playing less video games (*p* = 0.002).

As for parental control of time spent, the differences found are shown in Table 4. Differences were found in the type of game (*p* < 0.001–0.007) and the platform used (*p* < 0.001–0.019). The adolescents they monitored always played more on weekdays (*p* < 0.001) and weekends (*p* < 0.001); played online with friends (*p* < 0.001); and spent more money per month playing (*p* < 0.001).

Table 5 shows the differences according to academic performance. There were differences in the type of video game (*p* < 0.001–0.003) and the platforms used (*p* < 0.001). Adolescents with lower academic performance spent more hours gaming on weekdays (*p* < 0.001) and weekends (*p* < 0.001) and spent more money playing (*p* < 0.001). Adolescents with a higher academic performance played more offline (*p* < 0.001).

The differences according to the level of physical activity practiced are shown in Table 6. Differences were found in the type of video game played (*p* < 0.001–0.007) and the platform used (*p* < 0.001–0.035). The adolescents who practiced more physical activity played more video games during the weekend (*p* < 0.001); played more online with friends (*p* < 0.001); and spent between EUR 10 and 20 more than the rest (*p* = 0.036).

### 3.2. Differences in the Psychological Variables According to Gender

Table 7 shows the differences in the psychological variables according to the gender of the adolescents. Males showed a higher problematic use of video games (*p* < 0.001), higher gaming addiction (*p* < 0.001); higher scores in all categories in terms of gaming motives (*p* < 0.001), except for customization (*p* = 0.132); higher mood repair with the use of video games (*p* < 0.001) and higher scores in the passion scale (*p* < 0.001). Females showed higher scores in emotional symptoms (*p* < 0.001) and prosocial behavior (*p* < 0.001).

### 3.3. Differences in the Psychological Variables According to the Closest Environment

Table 8 shows the differences in the psychological variables according to whether the parents live together. It is observed that in those adolescents whose parents live together, mood repair was higher (*p* = 0.005). In the adolescents whose parents live apart, the score was higher in emotional symptoms (*p* < 0.001), conduct problems (*p* < 0.001) and peer relationship problems (*p* < 0.001).

It was also observed that adolescents whose parents always monitored them showed greater problematic use of video games (*p* < 0.001), greater gaming addiction (*p* < 0.001) as well as higher scores in all gaming motives (recreation: *p* = 0.003–0.048) and passion (*p* < 0.001). On the other hand, adolescents who were never monitored showed higher scores in competition (*p* = 0.022), violent reward (*p* < 0.001), emotional symptoms (*p* = 0.008) and prosocial behavior (*p* = 0.015); and lower scores in mood repair (*p* < 0.001) (Figure 1).

### 3.4. Differences in the Psychological Variables According to the Academic Performance of the Adolescents

The differences as a function of academic performance are found in Table 9. Adolescents who failed or passed with a low grade showed higher scores in problematic use of the video games (*p* < 0.001–0.035); gaming addiction (*p* < 0.001); passion (*p* < 0.001–0.034); emotional symptoms (*p* < 0.001); conduct problems (*p* < 0.001–0.002); and their main gaming motives were coping (*p* = 0.022–0.035), social interaction (*p* < 0.001–0.002) and violent reward (*p* < 0.001). They only showed lower scores than adolescents with good grades (A or B) in prosocial behavior (*p* < 0.001–0.003) (Figure 2).

### 3.5. Differences in the Psychological Variables According to the Level of Physical Activity

The differences in the psychological variables according to the level of physical activity of the adolescents are shown in Table 10. Adolescents with insufficient and light physical activity showed lower competitiveness (*p* < 0.001–0.022), cognitive development (*p* < 0.001–0.017), social interaction (*p* < 0.001–0.021) and violent reward (*p* < 0.001–0.005). In addition, these adolescents showed lower mood repair (*p* < 0.001–0.044), greater emotional symptoms (*p* < 0.001) and peer relationship problems (*p* < 0.001–0.026) as well as less hyperactivity/inattention (*p* < 0.001–0.026) (Figure 3).

### 3.6. Differences in the Psychological Variables According to the Game Form

Table 11 shows the differences according to game form. Problematic video game use, gaming addiction and passion for video games were higher in adolescents who played online as compared to those who played offline (*p* < 0.001–0.009). Regarding the motives for gaming, recreation (*p* < 0.001–0.002), competition (*p* < 0.001–0.009), cognitive development (*p* < 0.001), coping (*p* < 0.001–0.046), social interaction (*p* < 0.001) and violent reward (*p* < 0.001–0.007) were the main motives for gaming for those who played online. Mood repair was higher only when comparing adolescents who played online with friends compared to those who played alone offline (*p* = 0.045). And in emotional and behavioral symptoms, players who played online with strangers showed higher emotional symptoms (*p* < 0.001–0.014), higher conduct problems (*p* = 0.012) and peer relationship problems (*p* < 0.001) (Figure 4).

**Table 8 behavsci-14-01204-t008:** Differences in the psychological variables according to cohabitation of parents and parental control of time spent on video games.

	Cohabitation of Parents						Parental Control of Time Use						
Variable	Yes (*n* = 1848)	No (*n* = 665)	t	95% CI	d	*p*	Never (*n* = 1483)	Sometimes (*n* = 340)	Always (*n* = 690)	F	95% CI	ES	*p*
CERV													
Evasion	12.85 ± 4.30	12.86 ± 4.30	−0.05	−0.40; 0.38	0.04	0.960	14.12 ± 4.35	13.37 ± 3.82	14.87 ± 3.77	13.22	13.90; 14.34	0.02	<0.001
Negative consequences	12.35 ± 3.37	12.66 ± 3.79	−1.94	−0.62; 0.00	0.03	0.053	13.16 ± 3.65	13.05 ± 3.15	13.97 ± 3.34	11.48	13.21; 13.58	0.01	<0.001
Total	25.20 ± 7.31	25.52 ± 7.72	−0.94	−0.99; 0.35	0.07	0.351	27.28 ± 7.54	26.42 ± 6.57	28.84 ± 6.57	13.44	27.14; 27.89	0.02	<0.001
Gaming addiction	4.68 ± 4.78	4.95 ± 5.05	−1.20	−0.70; 0.17	0.05	0.234	5.72 ± 4.90	5.77 ± 4.82	7.07 ± 4.57	15.45	5.93; 6.44	0.02	<0.001
VMQ													
Recreation	9.66 ± 2.75	9.53 ± 2.88	0.87	−0.16; 0.42	0.03	0.390	9.54 ± 2.85	9.27 ± 2.91	9.95 ± 2.57	6.68	9.44; 9.73	0.01	0.001
Competition	7.41 ± 3.32	7.60 ± 3.30	−1.05	−0.53; 0.16	0.03	0.307	7.56 ± 3.36	6.94 ± 3.41	7.56 ± 3.16	3.91	7.18; 7.53	0.00	0.020
Cognitive development	4.82 ± 3.27	4.86 ± 3.26	−0.25	−0.39; 0.30	0.03	0.801	4.86 ± 3.35	4.37 ± 3.16	4.95 ± 3.12	3.08	4.56; 4.90	0.00	0.050
Coping	4.84 ± 3.43	4.91 ± 3.46	−0.42	−0.37; 0.24	0.03	0.685	5.42 ± 3.86	5.00 ± 3.87	5.88 ± 3.63	5.39	5.23; 5.63	0.01	0.005
Social interaction	5.01 ± 3.32	5.10 ± 3.49	−0.49	−0.44; 0.27	0.04	0.632	4.86 ± 3.44	4.77 ± 3.46	5.41 ± 3.13	5.77	4.84; 5.19	0.01	0.003
Violent reward	3.95 ± 3.66	4.23 ± 3.87	−1.40	−0.67; 0.11	0.04	0.166	4.29 ± 3.97	3.30 ± 3.49	3.88 ± 3.32	7.79	3.63; 4.02	0.01	<0.001
Customization	7.19 ± 3.73	6.85 ± 3.80	−1.71	−0.05; 0.74	0.04	0.093	6.94 ± 3.88	7.45 ± 3.75	7.17 ± 3.53	2.14	6.99; 7.39	0.00	0.124
Fantasy	6.07 ± 3.75	6.24 ± 3.80	−0.85	−0.57; 0.22	0.04	0.397	6.02 ± 3.82	5.88 ± 3.70	6.27 ± 3.71	1.19	5.86; 6.26	0.00	0.312
Mood repair	24.88 ± 7.13	23.96 ± 7.20	2.79	0.27; 1.56	0.71	0.005	24.08 ± 7.23	26.25 ± 7.33	25.38 ± 6.54	12.52	24.86; 25.61	0.01	<0.001
Passion scale													
Harmonious	8.58 ± 2.75	8.81 ± 2.93	−1.78	−0.48; 0.02	0.02	0.087	9.30 ± 2.95	8.93 ± 2.63	9.86 ± 2.67	12.10	9.21; 9.51	0.01	<0.001
Obsessive	9.23 ± 3.01	9.35 ± 3.15	−0.83	−0.39; 0.16	0.03	0.416	10.05 ± 3.05	9.62 ± 2.80	10.66 ± 2.85	13.38	9.95; 10.27	0.02	<0.001
SDQ													
Emotional symptoms	14.00 ± 4.77	14.95 ± 4.84	−4.33	−1.38; −0.52	0.48	<0.001	14.07 ± 4.86	13.07 ± 5.02	13.61 ± 4.57	5.01	13.33; 13.84	0.01	0.007
Conduct problems	12.19 ± 2.95	12.64 ± 3.09	−3.27	−0.72; −0.18	0.30	<0.001	12.50 ± 3.02	12.28 ± 3.21	12.41 ± 2.98	0.59	12.24; 12.56	0.00	0.565
Peer relationship problem	15.53 ± 2.51	15.87 ± 2.67	−2.88	−0.57; −0.11	0.26	<0.001	15.79 ± 2.61	15.41 ± 2.81	15.59 ± 2.44	2.70	15.46; 15.74	0.00	0.073
Hyperactivity/inattention	16.20 ± 2.97	16.29 ± 3.03	−0.68	−0.36; 0.17	0.03	0.496	16.21 ± 3.07	16.46 ± 3.46	16.37 ± 2.84	0.96	16.18; 16.51	0.00	0.384
Prosocial behavior	21.19 ± 3.15	20.96 ± 3.47	1.54	−0.06; 0.52	0.03	0.138	20.72 ± 3.45	21.38 ± 3.64	20.99 ± 3.10	4.23	20.85; 21.21	0.01	0.026

**Table 9 behavsci-14-01204-t009:** Differences in psychological variables according to academic performance.

Academic Performance									
Variable	F (*n* = 183)	D (*n* = 426)	C (*n* = 585)	B (*n* = 906)	A (*n* = 413)	F	95% CI	ES	*p*
CERV									
Evasion	15.08 ± 4.59	15.04 ± 4.16	14.37 ± 4.31	13.79 ± 3.88	13.75 ± 3.92	7.09	14.18; 14.63	0.02	<0.001
Negative consequences	14.62 ± 4.37	14.35 ± 3.89	13.55 ± 3.59	12.98 ± 3.14	12.40 ± 2.66	17.51	13.39; 13.77	0.04	<0.001
Total	29.70 ± 8.35	29.39 ± 7.58	27.92 ± 7.42	26.77 ± 6.56	26.15 ± 6.20	12.64	27.60; 28.37	0.03	<0.001
Gaming addiction	7.46 ± 5.56	7.30 ± 5.16	6.28 ± 4.86	5.59 ± 4.61	5.20 ± 3.99	11.46	6.11; 6.63	0.03	<0.001
VMQ									
Recreation	9.62 ± 2.76	9.72 ± 2.72	9.44 ± 2.90	9.69 ± 2.76	9.76 ± 2.67	0.79	9.49; 9.80	0.00	0.532
Competition	8.07 ± 3.37	7.56 ± 3.36	7.56 ± 3.36	7.28 ± 3.27	7.25 ± 3.26	1.98	7.36; 7.73	0.01	0.095
Cognitive development	4.86 ± 3.16	4.98 ± 3.33	4.79 ± 3.39	4.71 ± 3.14	4.89 ± 3.23	0.43	4.67; 5.02	0.00	0.786
Coping	5.94 ± 4.03	6.05 ± 3.82	5.53 ± 3.77	5.29 ± 3.70	5.05 ± 3.78	3.42	5.37; 5.78	0.01	0.009
Social interaction	5.84 ± 3.62	5.64 ± 3.35	5.21 ± 3.39	4.64 ± 3.27	4.49 ± 3.15	8.51	4.98; 5.35	0.02	<0.001
Violent reward	5.44 ± 3.98	4.65 ± 3.86	4.01 ± 3.70	3.67 ± 3.53	3.17 ± 3.60	11.47	3.99; 4.39	0.03	<0.001
Customization	6.74 ± 3.71	7.15 ± 3.81	6.78 ± 3.80	7.22 ± 3.77	7.41 ± 3.48	1.63	6.85; 7.26	0.00	0.163
Fantasy	6.23 ± 3.73	6.12 ± 3.76	5.87 ± 3.78	6.08 ± 3.78	6.22 ± 3.74	0.48	5.90; 6.31	0.00	0.754
Mood repair	23.41 ± 8.13	24.04 ± 7.25	24.93 ± 6.86	25.25 ± 7.01	25.47 ± 6.44	3.34	24.23; 25.00	0.01	0.100
Passion scale									
Harmonious	10.51 ± 3.53	10.15 ± 2.93	9.60 ± 2.89	8.96 ± 2.57	8.75 ± 2.43	18.36	9.44; 9.75	0.04	<0.001
Obsessive	10.77 ± 3.26	10.81 ± 3.24	10.24 ± 3.07	9.92 ± 2.78	9.65 ± 2.63	7.87	10.12; 10.44	0.02	<0.001
SDQ									
Emotional symptoms	14.77 ± 4.87	14.41 ± 4.71	14.01 ± 4.84	13.64 ± 4.79	12.44 ± 4.85	7.59	13.59; 14.12	0.02	<0.001
Conduct problems	13.44 ± 3.68	12.90 ± 3.16	12.58 ± 3.05	12.12 ± 2.81	11.76 ± 2.80	10.40	12.40; 12.72	0.02	<0.001
Peer relationship problem	16.06 ± 3.27	15.80 ± 2.75	15.87 ± 2.63	15.53 ± 2.40	15.46 ± 2.43	2.32	15.60; 15.88	0.01	0.055
Hyperactivity/inattention	16.72 ± 3.27	16.42 ± 3.23	16.49 ± 3.00	16.13 ± 3.03	15.94 ± 2.92	2.43	16.17; 16.51	0.01	0.056
Prosocial behavior	20.04 ± 3.66	20.24 ± 3.75	20.93 ± 3.27	21.22 ± 3.12	21.39 ± 3.27	7.88	20.58; 20.95	0.02	<0.001

**Table 10 behavsci-14-01204-t010:** Differences in psychological variables according to the level of physical activity of the adolescents.

Level of Physical Activity								
Variable	Insufficient (*n* = 540)	Light (*n* = 1028)	Moderate (*n* = 799)	Vigorous (*n* = 146)	F	95% CI	ES	*p*
CERV								
Evasion	14.07 ± 4.51	14.33 ± 4.05	14.33 ± 4.01	13.88 ± 4.10	0.67	13.91; 14.40	0.00	0.568
Negative consequences	13.27 ± 3.70	13.58 ± 3.52	13.32 ± 3.37	13.12 ± 3.51	1.15	13.11; 13.53	0.00	0.327
Total	27.34 ± 7.73	27.91 ± 7.06	27.65 ± 6.91	27.00 ± 7.28	0.84	27.05; 27.90	0.00	0.474
Gaming addiction	6.09 ± 5.13	6.25 ± 4.69	6.19 ± 4.85	5.82 ± 4.76	0.30	5.80; 6.38	0.00	0.823
VMQ								
Recreation	9.30 ± 3.00	9.53 ± 2.72	9.92 ± 2.70	9.81 ± 2.71	4.26	9.48; 9.81	0.01	0.052
Competition	6.52 ± 3.40	7.15 ± 3.27	8.19 ± 3.18	8.60 ± 2.94	26.17	7.42; 7.81	0.04	<0.001
Cognitive development	4.34 ± 3.38	4.70 ± 3.13	5.23 ± 3.31	4.99 ± 3.56	6.03	4.62; 5.01	0.01	<0.001
Coping	5.07 ± 3.77	5.59 ± 3.77	5.71 ± 3.78	5.16 ± 4.00	2.48	5.16; 5.61	0.00	0.059
Social interaction	4.23 ± 3.39	5.00 ± 3.34	5.47 ± 3.35	5.29 ± 3.08	10.09	4.80; 5.20	0.02	<0.001
Violent reward	3.40 ± 3.69	3.75 ± 3.61	4.54 ± 3.80	5.01 ± 3.74	11.00	3.96; 4.40	0.02	<0.001
Customization	7.25 ± 3.82	7.20 ± 3.64	7.07 ± 3.75	6.03 ± 4.11	3.49	6.66; 7.11	0.01	0.055
Fantasy	6.09 ± 3.79	6.17 ± 3.78	6.11 ± 3.73	5.45 ± 3.84	1.19	5.73; 6.18	0.00	0.311
Mood repair	22.76 ± 7.11	24.74 ± 6.93	25.76 ± 6.96	26.62 ± 7.09	15.69	24.55; 25.39	0.03	<0.001
Passion scale								
Harmonious	9.29 ± 2.90	9.54 ± 2.89	9.42 ± 2.74	9.01 ± 2.75	1.47	9.15; 9.49	0.00	0.220
Obsessive	9.96 ± 3.25	10.24 ± 2.89	10.25 ± 2.90	10.10 ± 3.11	0.84	9.96; 10.32	0.00	0.471
SDQ								
Emotional symptoms	15.11 ± 4.87	14.34 ± 4.67	12.73 ± 4.63	11.74 ± 4.76	29.57	13.20; 13.76	0.05	<0.001
Conduct problems	12.43 ± 3.05	12.48 ± 3.09	12.39 ± 2.93	12.46 ± 3.27	0.09	12.26; 12.63	0.00	0.964
Peer relationship problem	15.86 ± 2.63	15.89 ± 2.61	15.36 ± 2.48	15.38 ± 2.77	5.93	15.47; 15.78	0.01	<0.001
Hyperactivity/inattention	15.84 ± 2.95	16.35 ± 2.97	16.30 ± 3.14	17.23 ± 3.27	6.04	16.25; 16.61	0.01	<0.001
Prosocial behavior	20.48 ± 3.55	20.93 ± 3.33	21.09 ± 3.25	21.05 ± 3.65	2.43	20.69; 21.09	0.00	0.064

**Table 11 behavsci-14-01204-t011:** Differences in psychological variables according to the game form.

Game Form								
Variable	Online with Friends (*n* = 1393)	Online with Unknown (*n* = 290)	Offline Alone (*n* = 615)	Offline with Friends (*n* = 215)	F	95% CI	ES	*p*
CERV								
Evasion	14.70 ± 4.15	15.13 ± 4.09	12.93 ± 3.66	12.31 ± 3.47	26.67	13.47; 14.06	0.05	<0.001
Negative consequences	13.70 ± 3.55	14.29 ± 3.75	12.28 ± 2.82	12.46 ± 3.22	21.48	12.93; 13.43	0.04	<0.001
Total	28.40 ± 7.20	29.42 ± 7.25	25.21 ± 6.06	24.77 ± 6.33	27.42	26.44; 27.46	0.05	<0.001
Gaming addiction	6.50 ± 4.88	7.42 ± 4.88	4.82 ± 4.11	4.73 ± 4.70	17.96	5.52; 6.21	0.03	<0.001
VMQ								
Recreation	9.92 ± 2.58	9.79 ± 2.83	8.87 ± 3.05	9.38 ± 2.90	14.13	9.29; 9.69	0.02	<0.001
Competition	8.02 ± 3.18	7.96 ± 3.08	5.70 ± 3.10	6.59 ± 3.42	52.47	6.84; 7.30	0.09	<0.001
Cognitive development	5.05 ± 3.22	5.35 ± 3.15	4.02 ± 3.05	4.42 ± 3.63	11.31	4.48; 4.94	0.02	<0.001
Coping	5.82 ± 3.77	5.96 ± 3.70	4.59 ± 3.71	4.59 ± 3.85	12.15	4.97; 5.52	0.02	<0.001
Social interaction	5.98 ± 3.06	4.80 ± 3.39	2.50 ± 2.84	3.94 ± 2.61	125.26	4.08; 4.53	0.18	<0.001
Violent reward	4.40 ± 3.70	4.63 ± 3.76	2.67 ± 3.36	3.02 ± 3.39	24.67	3.42; 3.95	0.04	<0.001
Customization	6.95 ± 3.70	7.66 ± 3.59	7.27 ± 3.93	6.84 ± 4.05	2.16	6.90; 7.45	0.00	0.091
Fantasy	6.01 ± 3.75	6.67 ± 3.57	6.18 ± 3.91	5.60 ± 3.60	1.97	5.84; 6.39	0.00	0.117
Mood repair	25.23 ± 6.95	23.96 ± 7.31	24.09 ± 7.02	24.68 ± 7.35	3.35	23.98; 25.00	0.01	0.018
Passion scale								
Harmonious	9.70 ± 2.90	10.04 ± 2.80	8.55 ± 2.38	8.21 ± 2.46	23.66	8.92; 9.33	0.04	<0.001
Obsessive	10.50 ± 3.01	10.84 ± 3.07	9.16 ± 2.48	8.93 ± 2.43	27.68	9.64; 10.07	0.05	<0.001
SDQ								
Emotional symptoms	13.24 ± 4.71	15.39 ± 5.08	14.84 ± 4.86	13.41 ± 4.36	17.08	13.87; 14.57	0.03	<0.001
Conduct problems	12.45 ± 3.01	13.04 ± 3.33	12.15 ± 2.96	12.16 ± 3.05	3.43	12.23; 12.67	0.01	0.017
Peer relationship problem	15.43 ± 2.52	16.48 ± 2.82	16.00 ± 2.60	15.64 ± 2.12	10.81	15.70; 16.08	0.02	<0.001
Hyperactivity/inattention	16.31 ± 3.13	16.25 ± 2.87	16.28 ± 2.88	16.10 ± 3.12	0.13	16.01; 16.46	0.00	0.943
Prosocial behavior	20.98 ± 3.30	20.39 ± 3.85	21.04 ± 3.48	20.68 ± 2.85	1.73	20.52; 21.02	0.00	0.159

**Figure 1 behavsci-14-01204-f001:**
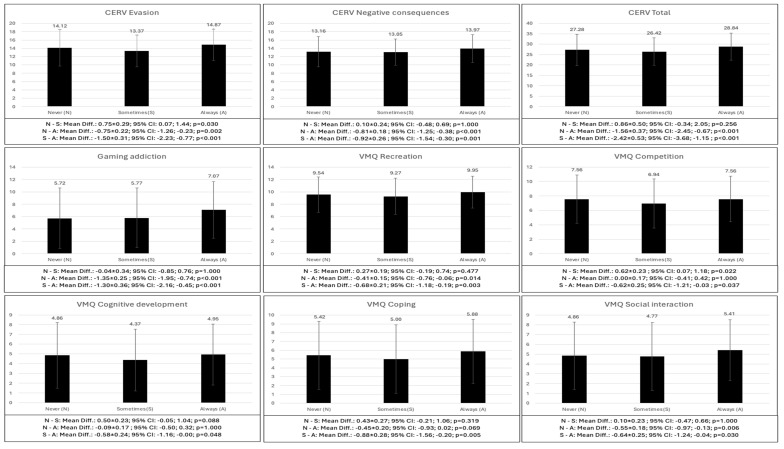
Bonferroni post hoc of the differences in the psychological variables according to the parental control of time spent.

**Figure 2 behavsci-14-01204-f002:**
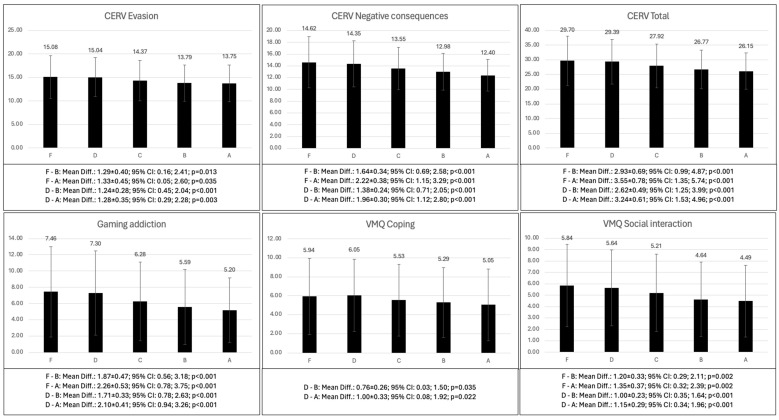
Bonferroni post hoc of the differences in the psychological variables according to academic performance (continued).

**Figure 3 behavsci-14-01204-f003:**
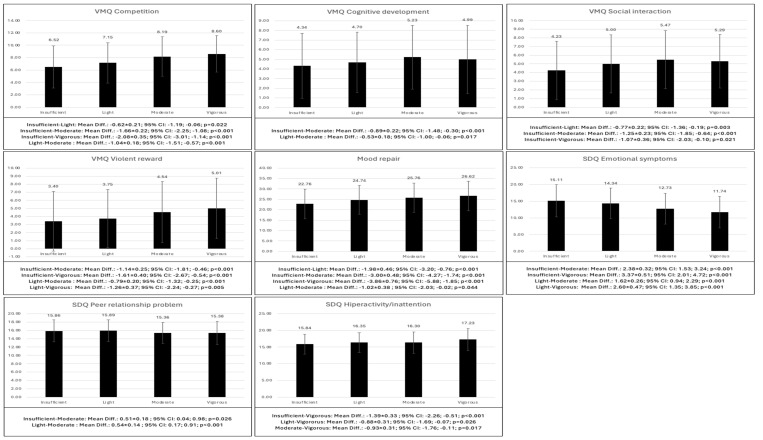
Bonferroni post hoc of the differences in the psychological variables according to the level of physical activity.

**Figure 4 behavsci-14-01204-f004:**
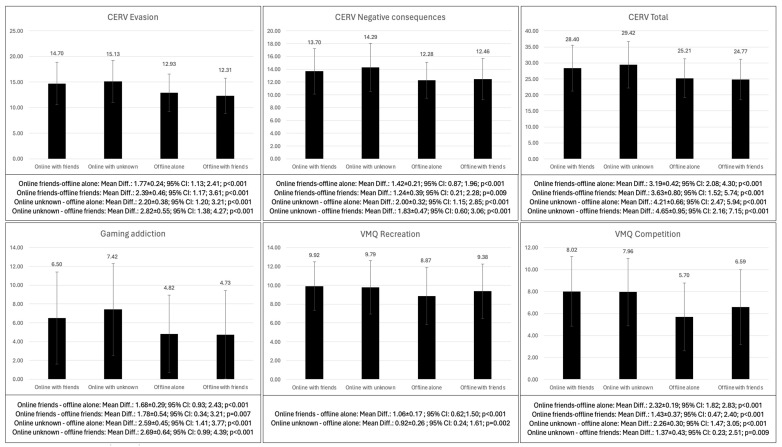
Bonferroni post hoc of the differences in the psychological variables according to the game form.

### 3.7. Relationship Between Psychological, Game-Related and Behavioral Variables

Correlational analysis between psychological variables and gaming variables, physical activity level and academic performance is shown in Table 12. It was found that physical activity level was positively related to gaming-related experiences (*p* < 0.001–0.004); gaming addiction (*p* = 0.004); gaming motives (*p* < 0.001–0.004); mood repair (*p* < 0.001); passion scale (*p* < 0.001–0.018); and most of the social, emotional and behavioral problems (*p* < 0.001–0.010). Academic performance was inversely related to gaming-related experiences (*p* < 0.001); gaming addiction (*p* < 0.001); all gaming motives (*p* < 0.001–0.016); the passion scale (*p* < 0.001); and social, emotional and behavioral problems (*p* < 0.001). Weekday and weekend play time correlated positively with all variables in both cases (*p* < 0.001–0.003) except for emotional symptoms (*p* < 0.001) and prosocial behavior (*p* < 0.001). Monthly money spent on video games correlated positively with all psychological variables (*p* < 0.001), with the exception of emotional symptoms (*p* = 0.002) and prosocial behavior (*p* < 0.001), with which it was inversely related.

Table 13 shows the linear regression analysis that identifies the game-related variables and the behavioral variables that could predict the psychological state. Although a predictor model was identified for all the variables, the explanatory capacity in some cases was very low. Therefore, it should be noted that gaming-related experiences, as well as gaming addiction, were predicted by time spent playing, money spent and academic performance. Regarding gaming motives, recreation, competition and coping were predicted by time spent, academic performance and physical activity level. Mood repair was predicted by physical activity level, academic performance and time spent on video games on the weekend. With respect to the passion scale, time spent, money spent and academic performance were found to be predictors. And attending to social, emotional and behavioral problems, academic performance, physical activity level, time and money spent on video games were predictors. 

## 4. Discussion

The first and second research questions that led to the present research were ‘What factors related to video game use (playing time, game type, platform) show differences according to gender, closest environment, academic performance and physical activity level in adolescents?’ and ‘How are psychological variables related with video game use (playing time, game type, platform), gender, closest environment, academic performance and physical activity level in adolescents?’ In order to answer both of these questions, the first aim was to determine the differences in the gaming variables and the psychological variables related to video games according to gender, the closest environment, academic performance and the level of physical activity.

### 4.1. Differences in the Gaming Variables According to Gender, Closest Environment, Academic Performance and Physical Activity Level

#### 4.1.1. Differences in Video Game Use According to Gender

It was observed that males played more fighting, strategy, role playing, shooting and sports games, while females preferred casual and simulation games. These results are similar to previous research, which showed that males preferred competitive games, while females preferred social and simulation games [7]. This could be due to several reasons: (a) the design and marketing of video games has historically been associated with gender [73]; (b) gender differences in gaming motivations which would lead to different game genre choices [9,74]; (c) the barriers female face in accessing the video game industry and the gaming community, as well as in participating in the historical design and advertising processes [75]; or (d) that male and female gaming patterns are influenced by gender norms, social expectations and gender stereotypes [76,77]. These gender differences make it necessary to implement policies and campaigns that promote equal participation of males and females in this field.

Similarly, males prefer to play online with friends, while females play offline alone. This could also be related to the type of games and the platform used, as females are less oriented towards games that feature competition and three-dimensional rotation [78]. On the other hand, the fact that males prefer playing online with friends may help to foster the positive reinforcement of social connections [79]. The relevance of these findings is that social games, understood as those played with multiple individuals, either cooperatively or competitively, increase well-being [43,80]; and non-social games, those that are single-player and lack the elements of human interaction that are present in social games, are related differently to player well-being [81]. Thus, video games with social elements have been shown to have similar positive effects on players’ social well-being [44,82]. Furthermore, socially oriented video games have been found to be associated with less problematic gaming symptoms compared to games with less or no social component [45].

#### 4.1.2. Differences in Video Game Use According to the Closest Environment

Regarding the cohabitation of parents and the parental control of time spent, the results showed a relationship between adolescents whose parents live apart and the time spent playing video games during the week. No previous research is known to have analyzed this question in the area of video games, although previous studies examined the use of other technologies, showing that adolescents whose parents live apart exhibit addictive behaviors towards the Internet or mobile phones, in a more prevalent manner than adolescents who live with both parents [83,84]. One possible explanation for these results is that at the time of parental separation, adolescents are affected mentally and behaviorally, which can lead to addictive or unhealthy behaviors on their part [85]. In addition, the separation of parents leads them to live alternatively in two households, with their respective set of rules, which increases the likelihood that certain rules will be relaxed [86], so adolescents may have more freedom to choose what they do.

In terms of parental control, the most important finding was that constant parental control was related to a higher use of video games, which could be due to an adverse reaction to strict control, where adolescents develop a more conflictual and obsessive relationship with video games as a form of resistance [87]. In contrast, those with more flexible parental control tended to exhibit more balanced behavior, suggesting that a less restrictive approach might be more effective in managing video game use [87]. In addition, the greatest parental control was found in adolescents who partook in more online gaming with friends, while the least parental control was found in those who played offline alone. This could be because playing offline alone is associated with less problematic and addictive video game use, as compared to playing online with friends, with players exhibiting more addictive behaviors [47], which could result in parents being more concerned about play time in the latter case, resulting in increased monitoring.

#### 4.1.3. Differences in Video Game Use According to Academic Performance

In terms of academic performance, the lowest-performing adolescents played more games during the week and on the weekend, mainly shooting and open-world games, on consoles and computers, and spent more money on games than the highest-performing adolescents. Regarding the day of play (weekday or weekend), studies on weekday play force the player to play at night, decreasing the time available for sleep [88], which is not the case when the player plays on weekends as they can better adjust their sleep hours [21,89]. Given that weekday play may lead to inadequate sleep compared to weekend play, and that a healthy circadian rhythm is crucial for well-being [90], it is likely that weekday play is worse for well-being, being associated with depressive and psychosomatic symptoms, which could affect academic performance [81,91]. Therefore, higher video game use may be associated with poorer academic performance, especially when playing on weekdays and, more specifically, before going to school [21]. This greater use of video games by adolescents with poorer academic performance may also reflect a compensatory use of video games as an escape mechanism from academic difficulties [92]. This behavior can contribute to a negative cycle, where increased play time interferes with academic performance, exacerbating school problems and reducing opportunities for educational success [22]. However, a noteworthy aspect of the results found is that adolescents with lower academic performance played shooting and open-world games to a greater extent, mainly on consoles and computers. No previous research is known that has analyzed different types of games or platforms in relation to academic performance, but these results may lead to future lines of research that analyze, to a greater extent, the type of video games and platforms played, and not only the time spent playing.

#### 4.1.4. Differences in Video Game Use According to Physical Activity Level

In terms of the level of physical activity, it was observed that adolescents who were less physically active played more role playing, casual, simulation and open-world games, while those who were more physically active played more shooting and sport and racing games. These results are partially similar to those from previous research, in which ball sport simulators (e.g., FIFA or NBA) were found to be associated with more physical activity [5]. This may be because the use of sports video games may be associated with increased participation in real-life sports in the adolescent population [93]. On the other hand, the fact that shooting games were associated with increased physical activity may be due to the fact that more aggressive adolescents tend to engage in physical sporting activities that allow them to reduce or mitigate aggressive behavior [94], and shooting games are classified as violent games that are also related to aggression [95]. However, future research is needed to directly analyze the relationship between the type of video game played and physical activity.

The more active adolescents played more online with friends, while the less active adolescents played offline alone. Previous research has shown that physical activity increases adolescents’ social relationships as they play mostly in teams and clubs, which is related to increased friendship attachment [96], and may also be extrapolated to other areas outside sport. Not surprisingly, previous research has pointed to the importance of the time spent by friends on both physical activity and the use of screens, and on the time spent on these two activities during adolescence [97].

In response to the first research question, the gaming variables that show significant differences are the type of platform and game used, as well as the form of gameplay in relation to gender; time spent playing and form of gameplay in relation to closest environment; time spent playing, platform and type of game and money spent in relation to academic performance; and type of game and form of gameplay in relation to level of physical activity.

### 4.2. Differences in the Psychological Variables According to Gender, Closest Environment, Academic Performance and Physical Activity Level

#### 4.2.1. Differences in Psychological Variables According to Gender

The results showed that males have a higher problematic video game use as compared to females, especially on the dimensions of avoidance and negative consequence, as well as a higher total video game addiction score. This suggests that males may be more susceptible to developing problem behaviors in relation to video games, which is consistent with previous studies that indicated that males tend to engage more in competitive and high-risk games, which could increase their vulnerability to these problems [98]. However, this appears to be influenced by the type of game played, with violent video games showing more controversial results, highlighting that women tend to show greater anxiety associated with these games, possibly due to their lower familiarity with the game mechanics [99].

It is also noteworthy that emotional repair through the use of video games was higher in males than in females. However, this type of emotional repair may lead to negative consequences, as it could perpetuate problematic video game use if it is used as the main coping mechanism for difficult situations [100], and even more so when considering that males have higher scores on the passion scale. This suggests that video games occupy a central place in their lives, not only as a form of leisure, but as an activity that defines their identity and sense of achievement [101]. However, obsessive passion is associated with negative effects on psychological well-being, such as increased anxiety and reduced satisfaction of basic psychological and life needs [102]. These findings reinforce the need to promote the balanced use of video games to maximize their benefits and minimize their risks [103], with adolescent males being a particularly sensitive population that should be monitored in this regard.

#### 4.2.2. Differences in the Psychological Variables According to the Closest Environment

In adolescents whose parents lived apart, mood repair was lower, while emotional and behavioral problems, specifically emotional symptoms, conduct problems and peer relationship problems, were larger, as compared to those whose parents lived together. These results suggest that a stable family environment may mitigate the negative effects associated with excessive video game use [41], through more effective emotional support, and helping adolescents to better manage stress and negative emotions, which in turn may reduce the need to turn to video games as an escape mechanism [104]. However, parents not living together was associated with adolescents showing more problematic video game use and greater emotional and behavioral problems, underlining the importance of the family context in adolescent mental health [105]. These findings suggest the need for interventions that address the family context as a key factor in adolescents’ mental health and their relationship with video games [106].

In terms of parental control of time spent playing video games, a relationship was observed between always monitored adolescents and increased problematic use, game addiction and passion for video games. This finding is in line with previous research that showed that greater parental control may be related to engagement in deceptive gaming behaviors [107], as a reactive response to perceived parental control, where adolescents develop a more conflictual relationship with video games as a form of resistance [34,87]. However, adolescents with greater parental control also showed higher scores on motives for playing. This is consistent with previous research that alludes to the use of video games as a pastime or entertainment, or for forgetting worries or relaxation in this population [108]. On the other hand, it is noteworthy that adolescents who did not experience any type of control showed lower scores on mood repair, as well as higher scores on emotional symptoms. Previous research has found that minimal parental control is necessary and beneficial for the emotional well-being of adolescents, without falling into overprotection or a total absence of control [87].

#### 4.2.3. Differences in the Psychological Variables According to Academic Performance

With regard to academic performance, lower-performing adolescents had higher problematic use and higher gaming addiction as compared to higher-performing adolescents, as well as greater passion for video games and greater emotional symptoms and behavioral problems. These results are consistent with previous research, in which addictive tendencies were negatively related to school performance [22]. One possible explanation for these results is that by spending too much time playing video games, adolescents may be reducing the time they spend on other activities such as studying or other educational activities [3]. Regarding the higher passion shown by adolescents with a lower academic performance, it is worth noting that the passion shown was obsessive. While harmonious passion could be beneficial and motivate adolescents to engage in activities in a balanced manner, obsessive passion is linked to an uncontrolled use of video games, which negatively affects their emotional and social well-being [98]. In addition, adolescents with a lower academic performance showed that one of their main motives for playing games was violent reward. This could be related to the fact that games that reward violent actions may increase aggressive behavior [109], which in turn is found to be related to lower academic performance [110].

#### 4.2.4. Differences in the Psychological Variables According to the Physical Activity Level

Another relevant finding of the present research was that there were no significant differences in problematic video game use and gaming addiction as a function of the level of physical activity practiced, which is similar to previous research, in which no relationships were found between internet and gaming addiction and the level of physical activity of high school students [111]. However, differences were found in play motives, with adolescents who engaged in less physical activity showing less competitiveness, cognitive development, social interaction and violent reward as compared to those who engaged in moderate and vigorous play. These results suggest that physical activity may be related to a more balanced and less problematic use of video games, which may contribute to better social and cognitive adjustment [112]. In addition, adolescents with lower levels of physical activity showed greater emotional symptoms and peer relationship problems. This could indicate that a low level of physical activity may be associated with an increased risk of emotional and social problems, which is consistent with previous studies that found an inverse relationship between physical activity, and symptoms of anxiety and depression in adolescents [113]. On the other hand, adolescents with higher levels of physical activity showed a greater capacity for mood repair through play. This reinforces the idea that physical activity is not only beneficial for physical health, but also for emotional well-being, by providing adolescents with more effective tools to manage stress [114].

#### 4.2.5. Differences in the Psychological Variables According to the Form of Play

In terms of the form of play, adolescents who play online, whether with friends or strangers, exhibit greater problematic video game use, with higher scores on avoidance and negative consequences, and a higher total score on the gaming addiction scale. These findings are consistent with previous research that suggested that online gaming, especially when performed with strangers, was associated with an increased risk of developing addictive behaviors and emotional problems [115]. However, these results are contrary to those found in previous research indicating that social games are associated with less problematic video game use compared to games that do not have a social component [45]. In terms of motives for playing video games, it was observed that adolescents who prefer to play online are mainly motivated by recreation, competition, cognitive development, coping, social interaction and violent reward-seeking. This contrasts with those who play offline, who show less motivation in these areas. Increased social interaction and competition are motives that are intrinsically linked to the satisfaction of emotional and social needs that may become dependent on the virtual environment in online gamers, which could be related to problematic video game use [116]. In addition, adolescents who play online also show greater emotional and behavioral problems, such as symptoms of anxiety and depression and difficulties in interpersonal relationships, as compared to those who play offline. This suggests that the online environment may exacerbate existing emotional difficulties or contribute to the development of new ones, given the anonymity and disinhibition that often characterize interaction in these environments [117].

In response to the second research question, significant differences were observed in problematic video game use, motives for playing, emotional repair and passion in relation to gender; in problematic video game use, game addiction, passion, mood repair, emotional and behavioral problems and problems in relationships with peers in relation to the closest environment; in problematic video game use, gaming addiction, passion and increased emotional and behavioral problems in relation to academic performance; in motives for gaming, mood repair and emotional and peer relationship problems according to physical activity level; and in problematic video game use, gaming addiction, motives for playing and emotional and behavioral problems according to the form of play.

### 4.3. Influence of Gaming Variables, Academic Performance and Physical Activity Practice on Psychological Variables

The third question posed for the research was ‘What are the factors that determine psychological variables in adolescents: gaming habits, physical activity or academic performance?’, and the second aim to respond to this was to determine which gaming variables and behavioral variables influence psychological variables in adolescents.

#### 4.3.1. Influence of Gaming Variables on Psychological Variables

As for the time spent playing video games during the week and on the weekend, both factors showed the strongest relationship with all psychological variables. In fact, the correlation was positive and moderate in most of the variables, except for the motives for playing and social, emotional and behavioral problems, where the correlation was weak and negative in some of the variables (emotional symptoms and prosocial behavior). It is worth noting that the correlation shown for weekend play time was higher than that shown for weekday play time, probably due to the fact that it is on weekends that adolescents play for longer periods of time [118]. Regarding the linear regression analysis, time spent playing during the weekdays and on the weekend were shown to be the most predictive variables in all cases, with the exception of social, emotional and behavioral problems, where they were not the most determinant variables in the model. What these results indicate is that gaming time is related to psychological problems associated with video games among adolescents, as is the case with other variables such as depression and stress, probably because longer exposure time makes players more vulnerable to video game addictions, but also to internet-related addictions [119,120].

Money spent on video games showed weak positive correlations with video game-related experiences, gaming addiction, motives for playing video games and passion. In this respect, linear regression analysis showed the predictive ability of money spent on video game-related experiences, on gaming addiction, on some of the gaming motives such as competitiveness, cognitive development, coping, social interaction or violent reward and on passion. These results follow the line of previous research in which the more time and money invested in video games, the worse the mental state of the players, affecting variables such as depression, anxiety, stress or aggressiveness [31,121]. The relevance of the results lies in the fact that the money spent can be considered as a factor that is related to the gaming experience of the players.

#### 4.3.2. Influence of Academic Performance on Psychological Variables

Academic performance correlated negatively with all psychological variables except mood repair and prosocial behavior. This seems to indicate that adolescents who perform better academically are those who have a better psychological state when playing video games, as they have less gaming addiction, less problematic use, lower scores on the passion scale and fewer social, emotional and behavioral problems. However, the correlations were weak between academic performance and these psychological variables analyzed. Regarding the linear regression analysis, the predictive capacity of academic performance on experiences related to video games, gaming addiction, gaming motives and passion is small; but on mood repair, emotional symptoms, behavioral problems and prosocial behavior, its influence is greater. These results add further evidence to previous research in which it was observed that addictive video game use could decrease academic performance [25], but the psychological state of video game use had not been analyzed with regard to the adolescent’s academic performance. Therefore, these results are relevant because they show a relationship between better academic performance and lower psychological consequences of playing video games.

#### 4.3.3. Influence of Physical Activity Practice on Psychological Variables

Physical activity level correlated positively, albeit weakly, with competitive play motives, cognitive development, social interaction and violent relationship, as well as with mood repair and hyperactivity/inattention, and correlated negatively with emotional symptoms. In fact, in the linear regression analysis, physical activity only appears as a predictor of the motives of recreation, competition, cognitive development, social interaction, violent reward and customization, as well as mood repair, emotional symptoms and hyperactivity/inattention, presenting a considerable influence on the model. These results add evidence to the existing scientific literature on the relationship between physical activity and psychological variables related to video games, as there seems to be a positive relationship between physical activity and motives to play and social, emotional and behavioral problems. This is highly relevant because it reinforces the beneficial role that video games can have on the psychological state of adolescents, since the previous literature had only shown benefits in those video games considered as active, but not in those played while sitting down [47], which are the ones analyzed in the present research.

In response to the third question, it was observed that there is a relationship between gaming habits, physical activity and academic performance and the psychological variables, with time spent playing video games being a variable that presents a high correlation with the psychological variables related to video games, although the level of physical activity, academic performance and money spent on video games should also be considered.

### 4.4. Limitations of the Present Study

Despite the novelty of the present research and the applicability of the results found, the present work is not without limitations. Firstly, the motivations for which participants decide to play video games have not been analyzed, which is a highly relevant aspect that may condition whether the behaviors with the game are adaptive or problematic, an aspect that the recent literature highlights as crucial for understanding their effects on well-being. Based on SDT [9], video games can have a positive impact on well-being when they satisfy the needs for autonomy, competence and social relatedness, being useful for coping with negative psychological states such as depression, suggesting that they may be promising for emotional regulation [12]; whereas when these needs are not met and the game is based on extrinsic or escapist motivations, the effects can be negative, reflected in higher anxiety, depression or life dissatisfaction [13,14]. Therefore, it is crucial to consider the motivations behind video game use, as psychological well-being can be positively or negatively affected [14,81,122]. Furthermore, video game use could displace other healthy activities such as exercise or meaningful social interactions, negatively affecting players’ well-being [123,124]. In view of the above, this is an important point for future research. Secondly, the sample size of the comparison groups is not homogeneous in all comparisons. Thirdly, the cross-sectional design of this study prevents us from analyzing the change over time of the variables analyzed and there is a possibility of reverse causation as an explanation of the results. Future research needs to look at these issues longitudinally to address this limitation. Fourthly, the measurement of physical activity, based on self-reports, introduces a possible response bias. Although the sample is representative of Galicia, extrapolation to other populations should be made with caution. And fifthly, although gender differences are analyzed, this study does not fully address how gender dynamics influence the relationship between video game use and psychological outcomes.

### 4.5. Theoretical and Practical Implications

In terms of the recommendations derived from the results found, this study makes significant contributions in both theoretical and practical terms, reinforcing the understanding of the factors that influence the use of video games among adolescents. From a theoretical perspective, the results extend knowledge about the role of variables such as gender, social context, academic performance and physical activity in the relationship between video game use and psychosocial well-being. These findings support and complement conceptual frameworks such as SDT [9,11] by showing how psychological motivations and needs, together with contextual factors, can be related with video game use. In this line, correlational and linear regression analysis allow us to establish that behavioral factors (physical activity and academic performance) and game-related variables (time spent playing video games and money spent) predict psychological variables related to video game use. Furthermore, one of the main findings that can be observed in the present research is that it is not only the problematic or addictive use of video games that should be considered in the adolescent population. As has been shown, there are other psychological constructs and variables that show differences according to factors such as gender, closest environment, academic performance or physical activity level. Moreover, a high percentage of adolescents play video games without presenting problematic or addictive use, so the results of the present study are highly relevant and offer the possibility of carrying out future research in this area.

From a practical perspective, the results offer guidance for designing intervention strategies aimed at adolescents. In this respect, the promotion of video games with social and cooperative components, or those that promote physical activity, could maximize the psychological and social benefits while minimizing the risks associated with problematic use. Furthermore, the identification of gender differences suggests the need for tailored approaches that consider the specific preferences and dynamics of each group. Also noteworthy is the fact that a longer time of use, whether on weekdays or weekends, seems to predict changes in psychological variables related to video games, which should be considered to avoid possible negative effects on health, regardless of the game used. Finally, these findings can serve as a basis for developing digital education and training campaigns aimed at parents, educators and adolescents to understand the benefits that can be gained from video game use and to steer them away from a negative view of video games. Such campaigns could also help to encourage a more balanced and reasonable use of video games, promoting healthy practices and minimizing associated risks. This comprehensive approach has the potential to positively influence the design of public policies and educational tools, contributing to improving the well-being of adolescents in the context of a growing video game culture.

## 5. Conclusions

Males play more video games than females, mainly fighting, strategy, role playing, shooting and sports games, on both console and PC. In addition, they have a worse addictive and emotional response to video games, and they also show more passion and different motives for playing video games. Adolescents who have separated parents and who have more parental control over the use of video games play more video games and present higher addiction values. Therefore, the optimal parental control seems to be one in which there is no exhaustive control, nor is this control omitted. Adolescents with poorer academic performance play more shooting and open-world games; they have a higher problematic and addictive use of video games, and their emotional response is also worse. Adolescents who play shooting and sport and racing games are more physically active; there is a difference according to the level of physical activity in the motives for playing and mood repair. Adolescents who played online games, mainly with strangers, showed more problematic and addictive use, and a worse emotional response. As predictor variables in the changes in psychological state related to video games, the hours of play stand out mainly, regardless of whether they are on weekdays or weekends, although the latter seem to be even more decisive. Money spent on video games, physical activity and academic performance, although relevant, are not as important predictors of these psychological variables.

## Figures and Tables

**Table 1 behavsci-14-01204-t001:** Socio-demographic characteristics of the study sample.

Variable	*n* (%)
Gender	
Males	1286 (51.20)
Females	1227 (48.80)
Educational level	
Compulsory secondary education	1825 (72.60)
Baccalaureate	450 (17.90)
Vocational studies	238 (9.50)
Physical activity level	
None	540 (21.50)
Light	1028 (41.00)
Moderate	799 (31.70)
Vigorous	146 (5.80)
Cohabitation of parents	
Yes	1848 (74.20)
No	665 (52.80)
Parental control of video game use	
Never	1483 (60.00)
Sometimes	340 (12.70)
Always	690 (27.20)
Academic performance	
F	183 (6.80)
D	426 (16.80)
C	585 (23.40)
B	906 (36.70)
A	413 (16.30)
Gaming form	
Online with friends	1393 (61.50)
Offline alone	615 (24.40)
Online with unknown	290 (8.90)
Offline with friends	215 (5.20)
Type of video game played	
Sport and racing	1025 (17.08)
Shooting	979 (16.32)
Open-world	967 (16.12)
Strategy	796 (13.27)
Simulation	584 (9.73)
Casual	550 (9.17)
Role Playing	503 (8.38)
Platforms	346 (5.77)
Fighting	250 (4.17)

**Table 2 behavsci-14-01204-t002:** Differences in the type of video games played, platform used, time spent, form of play and money spent per month according to gender.

Variable	Male (*n* = 1286)	Female (*n* = 1227)	Adj. Res	χ2; *p*	CC
Type of video game					
Platforms	168 (13.06)	176 (14.34)	1.7/−1.7	2.75; 0.098	0.034
Fighting	172 (13.37)	78 (6.36)	−5.3/5.3	28.10; <0.001	0.108
Strategy	487 (37.87)	298 (24.29)	−6.2/6.2	38.60; <0.001	0.126
Role Playing	295 (22.94)	207 (16.87	−2.9/2.9	8.70; 0.003	0.060
Shooting	785 (61.04)	186 (15.16)	−22.6/22.6	508.82; <0.001	0.419
Casual	161 (12.52)	386 (31.46)	12.6/−12.6	158.51; <0.001	0.249
Simulation	282 (21.93)	297 (24.21)	2.4/−2.4	5.68; 0.017	0.049
Open-world	625 (48.60)	334 (27.22)	−9.8/9.8	95.67; <0.001	0.196
Sport and racing	637 (49.53)	379 (30.89)	−8.2/8.2	68.27; <0.001	0.166
Platform					
Sony	663 (51.56)	215 (17.52)	−17.1/17.1	290.78; <0.001	0.328
PC	489 (38.02)	253 (20.62)	−8.7/8.7	76.40; <0.001	0.175
Nintendo	182 (14.15)	320 (26.08)	8.2/−8.2	68.00; <0.001	0.166
Tablet	79 (6.14)	163 (13.28)	6.6/−6.6	42.90; <0.001	0.132
Mobile Phone	504 (39.19)	714 (58.19)	11.1/−11.1	123.82; <0.001	0.221
Time spent (Weekdays)					
None	418 (32.50)	659 (53.71)	−11.1/11.1	220.32; <0.001	0.288
1 h	189 (14.70)	213 (17.36)	−1.9/1.9		
2 h	127 (9.88)	115 (9.37)	0.4/−0.4		
3 h	119 (9.25)	75 (6.11)	2.9/−2.9		
4 h	88 (6.84)	38 (3.10)	4.3/−4.3		
5 h	81 (6.30)	29 (2.36)	4.8/−4.8		
6 h	79 (6.14)	29 (2.36)	4.6/−4.6		
More than 6 h	148 (11.51)	26 (2.12)	9.2/−9.2		
Time spent (Weekend)					
None	100 (7.78)	492 (40.10)	−19.4/19.4	637.13; <0.001	0.454
1 h	104 (8.09)	204 (16.63)	−6.7/6.7		
2 h	117 (9.10)	153 (12.47)	−2.9/2.9		
3 h	152 (11.82)	98 (7.99)	3.1/−3.1		
4 h	137 (10.65)	80 (6.52)	3.6/−3.6		
5 h	131 (10.19)	56 (4.56)	5.3/−5.3		
6 h	151 (11.74)	41 (3.34)	7.8/−7.8		
More than 6 h	372 (28.93)	63 (5.13)	12.3/−12.3		
Game form					
Online with friends	849 (66.02)	431 (35.13)	12.1/−12.1	159.86; <0.001	0.267
Online with unknown	91 (7.08)	94 (7.66)	−1.9/1.9		
Offline alone	174 (13.53)	330 (26.89)	−11.1/11.1		
Offline with friends	49 (3.81)	60 (4.89)	−2.4/2.4		
Money spent (month)					
None	857 (66.64)	1046 (85.25)	−14.3/14.3	205.06; <0.001	0.281
<EUR 10	240 (18.66)	61 (4.97)	10.2/−10.2		
EUR 11–20	76 (5.91)	14 (1.14)	6.2/−6.2		
EUR 21–30	37 (2.88)	8 (0.65)	4.0/−4.0		
EUR 31–50	18 (1.40)	4 (0.33)	2.8/−2.8		
EUR 51–100	10 (0.78)	1 (0.08)	2.6/−2.6		
>EUR 100	12 (0.93)	2 (0.16)	2.5/−2.5		

CC: Contingency Coefficient.

**Table 3 behavsci-14-01204-t003:** Differences in the type of video games played, platform used, time spent, form of play and money spent per month according to the cohabitation of parents.

Variable	Yes (*n* = 1848)	No (*n* = 665)	Adj. Res	χ2; *p*	CC
Type of video game					
Platforms	250 (13.53)	93 (13.98)	0.5/−0.5	0.47; *p* = 0.793	0.014
Fighting	171 (9.25)	76 (11.43)	1.8/−1.8	3.49; *p* = 0.175	0.038
Strategy	562 (30.41)	226 (33.98)	2.1/−2.1	5.16; *p* = 0.076	0.046
Role playing	369 (19.97)	133 (20.00)	0.3/−0.3	0.38; *p* = 0.826	0.013
Shooting	721 (39.02)	250 (37.59)	−0.1/0.1	1.49; *p* = 0.475	0.025
Casual	392 (21.21)	154 (23.16)	1.3/−1.3	2.17; *p* = 0.339	0.030
Simulation	432 (23.38)	148 (22.26)	−0.3/0.3	0.39; *p* = 0.822	0.013
Open-world	730 (39.50)	229 (34.44)	−1.8/1.8	4.96; *p* = 0.084	0.046
Sport and racing	764 (41.34)	252 (37.89)	−1.1/1.1	2.87; *p* = 0.579	0.035
Platform					
Sony	641 (34.69)	238 (35.79)	1.1/−1.1	2.85; *p* = 0.241	0.034
PC	565 (30.57)	178 (26.77)	−1.4/1.4	2.37; *p* = 0.306	0.031
Nintendo	368 (19.91)	137 (20.60)	0.7/−0.7	0.81; *p* = 0.667	0.018
Tablet	199 (10.77)	45 (6.77)	−2.8/2.8	7.87; *p* = 0.020	0.057
Mobile Phone	863 (46.70)	352 (52.93)	3.5/−3.5	13.55; *p* = 0.001	0.075
Time spent (Weekdays)					
None	829 (44.86)	245 (36.84)	3.0/−3.0	36.62; *p* = 0.002	0.123
1 h	294 (15.91)	108 (16.24)	−0.5/0.5		
2 h	167 (9.04)	73 (10.98)	−1.7/1.7		
3 h	152 (8.23)	43 (6.47)	1.2/−1.2		
4 h	91 (4.92)	34 (5.11)	−0.5/0.5		
5 h	73 (3.95)	35 (5.26)	−1.6/1.6		
6 h	69 (3.73)	39 (5.86)	−2.5/2.5		
More than 6 h	128 (6.93)	49 (7.37)	−0.6/0.6		
Time spent (Weekend)					
None	439 (23.76)	147 (22.11)	0.4/−0.4	17.12; *p* = 0.378	0.084
1 h	217 (11.74)	89 (13.38)	−1.5/1.5		
2 h	202 (10.93)	65 (9.77)	0.5/−0.5		
3 h	188 (10.17)	61 (9.17)	0.3/−0.3		
4 h	173 (9.36)	47 (7.07)	1.5/−1.5		
5 h	145 (7.85)	42 (6.32)	1.0/−1.0		
6 h	142 (7.68)	51 (7.67)	−0.2/0.2		
More than 6 h	312 (16.88)	125 (18.80)	−1.6/1.6		
Game form					
Online with friends	958 (51.84)	319 (47.97)	1.3/−1.3	9.82; *p* = 0.278	0.069
Online with unknown	122 (6.60)	61 (9.17)	−2.4/2.4		
Offline alone	381 (20.62)	128 (19.25)	0.5/−0.5		
Offline with friends	78 (4.22)	31 (4.66)	−0.6/0.6		
Money spent (month)					
None	1409 (76.24)	482 (72.48)	1.0/−1.0	4.08; *p* = 0.982	0.041
<EUR 10	218 (11.80)	86 (12.93)	−1.0/1.0		
EUR 11–20	69 (3.73)	20 (3.01)	0.8/−0.8		
EUR 21–30	32 (1.73)	16 (2.41)	−1.2/1.2		
EUR 31–50	17 (0.92)	5 (0.75)	0.3/−0.3		
EUR 51–100	8 (0.43)	3 (0.45)	−0.1/0.1		
>EUR 100	9 (0.49)	5 (0.75)	−0.8/0.8		

CC: Contingency Coefficient.

**Table 4 behavsci-14-01204-t004:** Differences in the type of video games played, platform used, time spent, form of play and money spent per month according to the parental control.

Variable	Never (*n* = 1483)	Sometimes (*n* = 340)	Always (*n* = 690)	Adj. Res	χ2; *p*	CC
Type of video game						
Platforms	185 (12.47)	36 (10.59)	118 (17.10)	−3.1/3.1	9.92; *p* = 0.007	0.065
Fighting	133 (8.97)	33 (9.71)	78 (11.30)	1.6/−1.6	2.68; *p* = 0.262	0.034
Strategy	446 (30.07)	99 (29.12)	241 (34.93)	−2.1/2.1	4.56; *p* = 0.102	0.044
Role Playing	281 (18.95)	60 (17.65)	156 (22.61)	−2.0/2.0	3.85; *p* = 0.146	0.041
Shooting	502 (33.85)	140 (41.18)	330 (47.83)	6.2/−6.2	40.65; *p* < 0.001	0.132
Casual	355 (23.94)	59 (17.35)	127 (18.41)	−3.5/3.5	12.27; *p* = 0.002	0.072
Simulation	339 (22.86)	68 (20.00)	164 (23.77)	0.9/−0.9	0.87; *p* = 0.647	0.019
Open-world	530 (35.74)	124 (36.47)	308 (44.64)	−3.7/3.7	14.53; *p* < 0.001	0.079
Sport and racing	558 (37.63)	156 (45.88)	295 (42.75)	3.2/−3.2	14.58; *p* = 0.006	0.079
Platform						
Sony	454 (30.61)	135 (39.71)	287 (41.59)	−5.7/5.7	32.58; *p* < 0.001	0.118
PC	441 (29.74)	76 (22.35)	224 (32.46)	−2.6/2.6	7.92; *p* = 0.019	0.058
Nintendo	288 (19.42)	68 (20.00)	145 (21.01)	0.9/−0.9	0.90; *p* = 0.639	0.019
Tablet	120 (8.09)	55 (16.18)	65 (9.42)	4.9/−4.9	25.05; *p* < 0.001	0.103
Mobile Phone	726 (48.95)	151 (44.41)	310 (44.93)	−1.9/1.9	3.88; *p* = 0.144	0.041
Time spent (Weekdays)						
None	669 (45.11)	153 (45.00)	198 (28.70)	−7.3/7.3	95.60; *p* < 0.001	0.200
1 h	230 (15.51)	60 (17.65)	113 (16.38)	−1.4/1.4		
2 h	134 (9.04)	35 (10.29)	74 (10.72)	−1.7/1.7		
3 h	102 (6.88)	19 (5.59)	72 (10.43)	3.3/−3.3		
4 h	63 (4.25)	6 (1.76)	55 (7.97)	−2.7/2.7		
5 h	68 (4.59)	9 (2.65)	33 (4.78)	−1.5/1.5		
6 h	50 (3.37)	10 (2.94)	49 (7.10)	4.3/−4.3		
More than 6 h	113 (7.62)	12 (3.53)	49 (7.10)	−2.4/2.4		
Time spent (Weekend)						
None	455 (30.68)	39 (11.47)	38 (5.51)	13.6/−13.6	289.07; *p* < 0.001	0.347
1 h	211 (14.23)	45 (13.24)	50 (7.25)	−4.5/4.5		
2 h	156 (10.52)	49 (14.41)	65 (9.42)	2.8/−2.8		
3 h	109 (7.35)	46 (13.53)	95 (13.77)	−5.6/5.6		
4 h	100 (6.74)	35 (10.29)	83 (12.03)	−4.5/4.5		
5 h	78 (5.26)	31 (9.12)	77 (11.16)	−5.3/5.3		
6 h	101 (6.81)	15 (4.41)	77 (11.16)	4.2/−4.2		
More than 6 h	230 (15.51)	45 (13.24)	165 (23.91)	4.1/−4.1		
Game form						
Online with friends	673 (45.38)	189 (55.59)	415 (60.14)	−5.0/5.0	28.97; *p* < 0.001	0.118
Online with unknown	115 (7.75)	21 (6.18)	49 (7.10)	1.5/−1.5		
Offline alone	326 (21.98)	62 (18.24)	121 (17.54)	3.7/−3.7		
Offline with friends	70 (4.72)	17 (5.00)	20 (2.90)	−2.4/2.4		
Money spent (month)						
None	1163 (78.42)	244 (71.76)	467 (67.68)	−5.5/5.5	40.83; *p* < 0.001	0.132
<EUR 10	149 (10.05)	38 (11.18)	113 (16.38)	4.2/−4.2		
EUR 11–20	39 (2.63)	10 (2.94)	40 (5.80)	3.8/−3.8		
EUR 21–30	26 (1.75)	5 (1.47)	18 (2.61)	1.5/−1.5		
EUR 31–50	14 (0.94)	3 (0.88)	5 (0.72)	−0.5/0.5		
EUR 51–100	8 (0.54)	0 (0.00)	3 (0.43)	−1.3/1.3		
>EUR 100	9 (0.61)	2 (0.59)	3 (0.43)	−0.5/0.5		

CC: Contingency Coefficient.

**Table 5 behavsci-14-01204-t005:** Differences in the type of video games played, platform used, time spent, form of play and money spent per month according to academic performance.

Variable	F (*n* = 183)	D (*n* = 426)	C (*n* = 585)	B (*n* = 906)	A (*n* = 413)	Adj. Res	χ2; *p*	CC
Type of video game								
Platforms	29 (15.85)	62 (14.55)	86 (14.70)	101 (11.15)	61 (14.77)	2.9/−2.9	8.86; *p* = 0.065	0.062
Fighting	22 (12.02)	44 (10.33)	61 (10.43)	87 (9.60)	27 (6.54)	2.1/−2.1	6.43; *p* = 0.169	0.053
Strategy	46 (25.14)	122 (28.64)	184 (31.45)	298 (32.89)	114 (27.60)	1.6/−1.6	4.30; *p* = 0.367	0.043
Role Playing	35 (19.13)	93 (21.83)	114 (19.49)	160 (17.66)	79 (19.13)	−1.8/1.8	4.59; *p* = 0.332	0.044
Shooting	74 (40.44)	174 (40.85)	238 (40.68)	351 (38.74)	105 (25.42)	−5.2/5.2	29.95; *p* < 0.001	0.114
Casual	25 (13.66)	76 (17.84)	114 (19.49)	222 (24.50)	100 (24.21)	2.5/−2.5	16.71; *p* = 0.002	0.085
Simulation	49 (26.78)	94 (22.07)	128 (21.88)	198 (21.85)	88 (21.31)	2.0/−2.0	4.05; *p* = 0.399	0.042
Open-world	71 (38.80)	176 (41.31)	207 (35.38)	355 (39.18)	120 (29.06)	3.3/−3.3	16.40; *p* = 0.003	0.084
Sport and racing	72 (39.34)	166 (38.97)	237 (40.51)	353 (38.96)	156 (37.77)	−0.8;0.8	4.25; *p* = 0.834	0.043
Platform								
Sony	73 (38.89)	160 (37.56)	207 (35.38)	299 (33.00)	100 (24.21)	3.9/−3.9	24.30; *p* < 0.001	0.102
PC	65 (35.52)	118 (27.70)	182 (31.11)	258 (28.48)	83 (20.10)	−3.6/3.6	21.34; *p* < 0.001	0.095
Nintendo	21 (11.48)	67 (15.73)	94 (16.07)	206 (22.74)	89 (21.55)	3.2/−3.2	21.22; *p* < 0.001	0.095
Tablet	12 (6.56)	23 (5.40)	45 (7.69)	99 (10.93)	54 (13.08)	3.2/−3.2	21.33; *p* < 0.001	0.096
Mobile Phone	90 (49.18)	198 (46.48)	277 (47.35)	418 (46.14)	188 (45.52)	1.6/−1.6	3.24; *p* = 0.519	0.037
Time of use (Weekdays)								
None	52 (28.42)	170 (39.91)	219 (37.44)	401 (44.26)	206 (49.88)	4.3/−4.3	75.55; *p* < 0.001	0.179
1 h	24 (13.11)	52 (12.21)	103 (17.61)	156 (17.22)	53 (12.83)	−2.0/2.0		
2 h	18 (9.84)	38 (8.92)	55 (9.40)	80 (8.83)	41 (9.93)	−0.8/0.8		
3 h	13 (7.10)	31 (7.28)	53 (9.06)	67 (7.40)	22 (5.33)	1.7/−1.7		
4 h	9 (4.92)	27 (6.34)	23 (3.93)	51 (5.63)	10 (2.42)	−2.4/2.4		
5 h	8 (4.37)	25 (5.87)	30 (5.13)	29 (3.20)	17 (4.12)	−2.2/2.2		
6 h	13 (7.10)	22 (5.16)	25 (4.27)	34 (3.75)	12 (2.91)	2.3/−2.3		
More than 6 h	22 (12.02)	34 (7.98)	46 (7.86)	48 (5.30)	18 (4.36)	3.4/−3.4		
Time of use (Weekend)								
None	34 (18.58)	80 (18.78)	128 (21.88)	194 (21.41)	136 (32.93)	5.8/−5.8	91.28; *p* < 0.001	0.196
1 h	10 (5.46)	33 (7.75)	77 (13.16)	136 (15.01)	43 (10.41)	3.4/−3.4		
2 h	22 (12.02)	45 (10.56)	57 (9.74)	94 (10.38)	46 (11.14)	1.0/−1.0		
3 h	17 (9.29)	35 (8.22)	60 (10.26)	92 (10.15)	30 (7.26)	−1.4/1.4		
4 h	12 (6.56)	33 (7.75)	53 (9.06)	82 (9.05)	32 (7.75)	0.7/−0.7		
5 h	11 (6.01)	37 (8.69)	38 (6.50)	72 (7.95)	21 (5.08)	−1.6/1.6		
6 h	14 (7.65)	35 (8.22)	52 (8.89)	68 (7.51)	23 (5.57)	−1.6/1.6		
More than 6 h	42 (22.95)	101 (23.71)	98 (16.75)	134 (14.79)	49 (11.86)	2.2/−2.2		
Game form								
Online with friends	91 (49.73)	216 (50.70)	303 (51.70)	459 (50.66)	168 (40.68)	−2.7/2.7	41.18; *p* < 0.001	0.143
Online with unknown	13 (7.10)	45 (10.56)	44 (7.52)	54 (5.96)	28 (6.78)	2.7/−2.7		
Offline alone	33 (18.03)	72 (16.90)	113 (19.32)	189 (20.86)	94 (22.76)	2.5/−2.5		
Offline with friends	1 (0.55)	15 (3.52)	14 (2.39)	54 (5.96)	21 (5.08)	3.1/−3.1		
Money spent (month)								
None	112 (61.20)	300 (70.42)	434 (74.19)	684 (75.50)	315 (76.27)	−2.6/2.6	54.01; *p* < 0.001	0.153
<EUR 10	23 (12.57)	54 (12.68)	66 (11.28)	98 (10.82)	48 (11.62)	−1.0/1.0		
EUR 11–20	6 (3.28)	16 (3.76)	24 (4.10)	36 (3.97)	5 (1.21)	−2.7/2.7		
EUR 21–30	8 (4.37)	5 (1.17)	14 (2.39)	17 (1.88)	3 (0.73)	2.8/−2.8		
EUR 31–50	2 (1.09)	8 (1.88)	5 (0.85)	5 (0.55)	1 (0.24)	2.6/−2.6		
EUR 51–100	2 (1.09)	6 (1.41)	0 (0.00)	3 (0.33)	0 (0.00)	3.3/−3.3		
>EUR 100	3 (1.64)	2 (0.47)	3 (0.51)	4 (0.44)	1 (0.24)	2.4/−2.4		

CC: Contingency Coefficient.

**Table 6 behavsci-14-01204-t006:** Differences in the type of video games played, platform used, time spent, form of play and money spent per month according to the physical activity level.

Variable	Insufficient (*n* = 540)	Light (*n* = 1028)	Moderate (*n* = 799)	Vigorous (*n* = 146)	Adj. Res	χ2; *p*	CC
Type of video game							
Platforms	76 (14.07)	156 (15.18)	91 (11.39)	21 (14.38)	−2.3/2.3	5.65; *p* = 0.130	0.048
Fighting	44 (8.15)	109 (10.60)	78 (9.76)	18 (12.33)	−1.5/1.5	3.00; *p* = 0.392	0.035
Strategy	174 (32.22)	317 (30.84)	258 (32.29)	46 (31.51)	−0.8/0.8	0.784; *p* = 0.853	0.018
Role Playing	116 (21.48)	230 (22.37)	129 (16.15)	28 (19.18)	−3.3/3.3	12.18; *p* = 0.007	0.071
Shooting	147 (27.22)	391 (38.04)	354 (44.31)	83 (56.85)	4.4/−4.4	57.99; *p* < 0.001	0.155
Casual	153 (28.33)	226 (21.98)	152 (19.02)	18 (12.33)	4.4/−4.4	27.29; *p* < 0.001	0.106
Simulation	139 (25.74)	259 (25.19)	158 (19.77)	26 (17.81)	−2.8/2.8	13.03; *p* = 0.005	0.074
Open-world	205 (37.96)	430 (41.83)	293 (36.67)	37 (25.34)	−3.6/3.6	18.02; *p* < 0.001	0.086
Sport and racing	141 (26.11)	372 (36.19)	421 (52.69)	89 (60.96)	8.4/−8.4	144.00; *p* < 0.001	0.244
Platform							
Sony	121 (22.41)	340 (33.07)	346 (43.30)	77 (52.74)	−6.9/6.9	82.80; *p* < 0.001	0.185
PC	150 (27.78)	324 (31.52)	234 (29.29)	39 (26.71)	1.5/−1.5	2.97; *p* = 0.397	0.035
Nintendo	117 (21.67)	228 (22.18)	141 (17.65)	19 (13.01)	−2.3/2.3	11.50; *p* = 0.009	0.069
Tablet	41 (7.59)	108 (10.51)	85 (10.64)	9 (6.16)	−1.8/1.8	6.29; *p* = 0.098	0.051
Mobile Phone	276 (51.11)	520 (50.58)	356 (44.56)	70 (47.95)	−2.8/2.8	8.62; *p* = 0.035	0.060
Time of use (Weekdays)							
None	249 (46.11)	419 (40.76)	348 (43.55)	68 (46.58)	−2.0/2.0	24.73; *p* = 0.420	0.101
1 h	87 (16.11)	179 (17.41)	111 (13.89)	27 (18.49)	−2.0/2.0		
2 h	52 (9.63)	94 (9.14)	78 (9.76)	18 (12.33)	1.1/−1.1		
3 h	34 (6.30)	82 (7.98)	69 (8.64)	10 (6.85)	−1.5/1.5		
4 h	25 (4.63)	46 (4.47)	48 (6.01)	7 (4.79)	1.6/−1.6		
5 h	19 (3.52)	51 (4.96)	37 (4.63)	3 (2.05)	−1.4/1.4		
6 h	27 (5.00)	50 (4.86)	28 (3.50)	4 (2.74)	−1.4/1.4		
More than 6 h	35 (6.48)	79 (7.68)	55 (6.88)	6 (4.11)	−1.4/1.4		
Time of use (Weekend)							
None	188 (34.81)	218 (21.21)	158 (19.77)	30 (20.55)	6.9/−6.9	80.63; *p* < 0.001	0.181
1 h	62 (11.48)	133 (12.94)	91 (11.39)	19 (13.01)	1.1/−1.1		
2 h	51 (9.44)	130 (12.65)	73 (9.14)	17 (11.64)	2.6/−2.6		
3 h	56 (10.37)	91 (8.85)	91 (11.39)	13 (8.90)	1.6/−1.6		
4 h	40 (7.41)	84 (8.17)	80 (10.01)	15 (10.27)	1.5/−1.5		
5 h	23 (4.26)	78 (7.59)	74 (9.26)	13 (8.90)	−3.2/3.2		
6 h	41 (7.59)	75 (7.30)	69 (8.64)	10 (6.85)	1.1/−1.1		
More than 6 h	24 (4.44)	85 (8.27)	70 (8.76)	18 (12.33)	−3.3/3.3		
Game form							
Online with friends	214 (39.63)	501 (48.74)	467 (58.45)	105 (71.92)	5.6/−5.6	90.11; *p* < 0.001	0.207
Online with unknown	38 (7.04)	101 (9.82)	42 (5.26)	5 (3.42)	3.7/−3.7		
Offline alone	151 (27.96)	220 (21.40)	118 (14.77)	20 (13.70)	6.0/−6.0		
Offline with friends	23 (4.26)	48 (4.67)	38 (4.76)	1 (0.68)	−2.4/2.4		
Money spent (month)							
None	439 (81.30)	768 (74.71)	591 (73.97)	111 (76.03)	3.3/−3.3	30.19; *p* = 0.036	0.112
<EUR 10	51 (9.44)	134 (13.04)	101 (12.64)	17 (11.64)	−2.2/2.2		
EUR 11–20	13 (2.41)	31 (3.02)	36 (4.51)	10 (6.85)	2.2/−2.2		
EUR 21–30	4 (0.74)	24 (2.33)	20 (2.50)	1 (0.68)	−2.3/2.3		
EUR 31–50	3 (0.56)	13 (1.26)	6 (0.75)	0 (0.00)	1.8/−1.8		
EUR 51–100	3 (0.56)	4 (0.39)	3 (0.38)	1 (0.68)	0.5/−0.5		
>EUR 100	5 (0.93)	3 (0.29)	5 (0.63)	1 (0.68)	−1.5/1.5		

CC: Contingency Coefficient.

**Table 7 behavsci-14-01204-t007:** Differences in the psychological variables according to gender.

Variable	Male (*n* = 1286)	Female (*n* = 1227)	t	95% CI	d	*p*
CERV						
Evasion	14.68 ± 4.16	10.92 ± 3.54	24.21	3.45; 4.06	0.39	<0.001
Negative consequences	13.74 ± 3.58	11.03 ± 2.77	21.05	2.46; 2.96	0.32	<0.001
Total	28.42 ± 7.26	21.95 ± 6.02	24.14	5.94; 6.99	0.67	<0.001
Gaming addiction	6.49 ± 4.79	2.87 ± 4.16	20.09	3.27;3.98	0.45	<0.001
*VMQ*						
Recreation	10.13 ± 2.52	8.73 ± 2.98	10.72	1.15; 1.67	0.27	<0.001
Competition	8.26 ± 3.08	6.03 ± 3.23	14.54	1.92; 2.52	0.31	<0.001
Cognitive development	5.52 ± 3.24	3.58 ± 2.91	12.79	1.65; 2.25	0.31	<0.001
Coping	6.01 ± 3.65	3.61 ± 2.68	18.62	2.15; 2.65	0.32	<0.001
Social interaction	5.81 ± 3.25	3.60 ± 3.07	14.28	1.91; 2.52	0.32	<0.001
Violent reward	4.87 ± 3.70	2.47 ± 3.24	13.94	2.07; 2.74	0.35	<0.001
Customization	7.20 ± 3.73	6.92 ± 3.77	1.53	−0.08; 0.64	0.04	0.132
Fantasy	6.45 ± 3.66	5.48 ± 3.86	5.30	0.61; 1.32	0.37	<0.001
Mood repair	25.77 ± 7.00	23.48 ± 7.11	8.10	1.74; 2.85	0.71	<0.001
Passion scale						
Harmonious	9.71 ± 2.91	7.50 ± 2.18	21.43	2.01; 2.41	0.26	<0.001
Obsessive	10.49 ± 3.02	7.96 ± 2.49	22.71	2.30; 2.74	0.28	<0.001
SDQ						
Emotional symptoms	12.43 ± 4.23	16.15 ± 4.63	−20.99	−4.07; −3.37	0.44	<0.001
Conduct problems	12.33 ± 3.01	12.27 ± 2.98	0.55	−0.17; 0.30	0.03	0.581
Peer relationship problem	15.57 ± 2.57	15.65 ± 2.56	−0.77	−0.28; 0.12	0.02	0.440
Hyperactivity/inattention	16.30 ± 3.13	16.16 ± 2.81	1.20	−0.09; 0.38	0.03	0.234
Prosocial behavior	20.63 ± 3.33	21.66 ± 3.05	−8.03	−1.28; −0.78	0.32	<0.001

**Table 12 behavsci-14-01204-t012:** Correlational analysis between psychological variables and gaming variables, physical activity level and academic performance.

	Physical Activity Level	Academic Performance	Time Spent (Weekdays)	Time Spent (Weekend)	Money Spent (Month)
CERV—Evasion	0.067; *p* < 0.001	−0.152; *p* < 0.001	0.546; *p* < 0.001	0.709; *p* < 0.001	0.354; *p* < 0.001
CERV—Negative consequences	0.057; *p* = 0.004	−0.194; *p* < 0.001	0.489; *p* < 0.001	0.640; *p* < 0.001	0.361; *p* < 0.001
CERV—Total	0.066; *p* = 0.001	−0.179; *p* < 0.001	0.546; *p* < 0.001	0.711; *p* < 0.001	0.376; *p* < 0.001
Gaming addiction	0.057; *p* = 0.004	−0.170; *p* < 0.001	0.531; *p* < 0.001	0.666; *p* < 0.001	0.352; *p* < 0.001
VMQ—Recreation	0.078; *p* < 0.001	0.013; *p* = 0.591	0.238; *p* < 0.001	0.464; *p* < 0.001	0.177; *p* < 0.001
VMQ—Competition	0.207; *p* < 0.001	−0.058; *p* = 0.016	0.267; *p* < 0.001	0.395; *p* < 0.001	0.217; *p* < 0.001
VMQ—Cognitive development	0.103; *p* < 0.001	−0.012; *p* = 0.616	0.275; *p* < 0.001	0.401; *p* < 0.001	0.214; *p* < 0.001
VMQ—Coping	0.058; *p* = 0.004	−0.102; *p* < 0.001	0.380; *p* < 0.001	0.535; *p* < 0.001	0.286; *p* < 0.001
VMQ—Social interaction	0.114; *p* < 0.001	−0.133; *p* < 0.001	0.295; *p* < 0.001	0.496; *p* < 0.001	0.294; *p* < 0.001
VMQ—Violent reward	0.146; *p* < 0.001	−0.158; *p* < 0.001	0.244; *p* < 0.001	0.347; *p* < 0.001	0.237; *p* < 0.001
VMQ—Customization	−0.055; *p* = 0.019	0.040; *p* = 0.093	0.188; *p* < 0.001	0.226; *p* < 0.001	0.092; *p* < 0.001
VMQ—Fantasy	−0.023; *p* = 0.316	0.001; *p* = 0.952	0.218; *p* < 0.001	0.314; *p* < 0.001	0.143; *p* < 0.001
Mood repair	0.157; *p* < 0.001	0.101; *p* < 0.001	0.029; *p* = 0.145	0.083; *p* < 0.001	0.034; *p* = 0.100
Passion scale—Harmonious	0.047; *p* = 0.018	−0.205; *p* < 0.001	0.510; *p* < 0.001	0.649; *p* < 0.001	0.336; *p* < 0.001
Passion scale—Obsessive	0.075; *p* < 0.001	−0.154; *p* < 0.001	0.520; *p* < 0.001	0.673; *p* < 0.001	0.348; *p* < 0.001
SDQ—Emotional symptoms	−0.206; *p* < 0.001	−0.093; *p* < 0.001	−0.088; *p* < 0.001	−0.187; *p* < 0.001	−0.064; *p* = 0.002
SDQ—Conduct problems	0.006; *p* = 0.751	−0.148; *p* < 0.001	0.078; *p* < 0.001	0.079; *p* < 0.001	0.071; *p* < 0.001
SDQ—Peer relationship problems	−0.049; *p* = 0.014	−0.072; *p* < 0.001	0.060; *p* = 0.003	0.008; *p* = 0.709	−0.016; *p* = 0.430
SDQ—Hyperactivity/inattention	0.104; *p* < 0.001	−0.069; *p* < 0.001	−0.008; *p* = 0.707	−0.007; *p* = 0.713	−0.010; *p* = 0.617
SDQ—Prosocial behavior	0.052; *p* = 0.010	0.133; *p* < 0.001	−0.150; *p* < 0.001	−0.151; *p* < 0.001	−0.145; *p* < 0.001

**Table 13 behavsci-14-01204-t013:** Linear regression analysis of the psychological variables depends on gaming variables, physical activity level and academic performance.

Analysis	R^2^	*p*	Independent Variables Included	Standardized Coefficients (B)	*p*
CERV—Evasion					
Model 1	0.519	<0.001	Time spent (weekend)	0.565	<0.001
			Time spent (weekdays)	0.157	<0.001
			Money spent (month)	0.089	<0.001
			Academic performance	−0.038	0.010
CERV—Negative consequences					
Model 1	0.441	<0.001	Time spent (weekend)	0.496	<0.001
			Money spent (month)	0.124	<0.001
			Time spent (weekdays)	0.127	<0.001
			Academic performance	−0.094	<0.001
CERV—Total					
Model 1	0.532	<0.001	Time spent (weekend)	0.562	<0.001
			Time spent (weekdays)	0.151	<0.001
			Money spent (month)	0.110	<0.001
			Academic performance	−0.067	<0.001
Gaming addiction					
Model 1	0.471	<0.001	Time spent (weekend)	0.502	<0.001
			Time spent (weekdays)	0.176	<0.001
			Money spent (month)	0.099	<0.001
			Academic performance	−0.069	<0.001
VMQ—Recreation					
Model 1	0.228	<0.001	Time spent (weekend)	0.474	<0.001
			Academic performance	0.070	0.001
			Physical activity level	0.049	0.023
VMQ—Competition					
Model 1	0.207	<0.001	Time spent (weekend)	0.294	<0.001
			Physical activity level	0.203	<0.001
			Time spent (weekdays)	0.113	<0.001
			Money spent (month)	0.087	<0.001
VMQ—Cognitive development					
Model 1	0.185	<0.001	Time spent (weekend)	0.321	<0.001
			Time spent (weekdays)	0.107	<0.001
			Physical activity level	0.089	<0.001
			Money spent (month)	0.083	<0.001
VMQ—Coping					
Model 1	0.299	<0.001	Time spent (weekend)	0.457	<0.001
			Money spent (month)	0.100	<0.001
			Time spent (weekdays)	0.061	0.007
VMQ—Social interaction					
Model 1	0.277	<0.001	Time spent (weekend)	0.402	<0.001
			Money spent (month)	0.137	<0.001
			Physical activity level	0.107	<0.001
			Time spent (weekdays)	0.058	0.017
			Academic performance	−0.082	<0.001
VMQ—Violent reward					
Model 1	0.174	<0.001	Time spent (weekend)	0.239	<0.001
			Physical activity level	0.144	<0.001
			Money spent (month)	0.128	<0.001
			Time spent (weekdays)	0.089	<0.001
			Academic performance	−0.124	<0.001
VMQ—Customization					
Model 1	0.070	<0.001	Time spent (weekend)	0.189	<0.001
			Time spent (weekdays)	0.100	<0.001
			Academic performance	0.081	<0.001
			Physical activity level	−0.069	0.004
VMQ—Fantasy					
Model 1	0.110	<0.001	Time spent (weekend)	0.271	<0.001
			Time spent (weekdays)	0.081	0.003
			Academic performance	0.052	0.025
Mood repair					
Model 1	0.040	<0.001	Physical activity level	0.141	<0.001
			Academic performance	0.095	<0.001
			Time spent (weekend)	0.093	<0.001
Passion scale—Harmonious					
Model 1	0.451	<0.001	Time spent (weekend)	0.500	<0.001
			Time spent (weekdays)	0.156	<0.001
			Money spent (month)	0.086	<0.001
			Academic performance	−0.102	<0.001
Passion scale—Obsessive					
Model 1	0.472	<0.001	Time spent (weekend)	0.527	<0.001
			Time spent (weekdays)	0.153	<0.001
			Money spent (month)	0.097	<0.001
			Academic performance	−0.047	0.003
SDQ—Emotional symptoms					
Model 1	0.077	<0.001	Academic performance	−0.102	<0.001
			Physical activity level	−0.174	<0.001
			Time spent (weekend)	−0.188	<0.001
SDQ—Conduct problems					
Model 1	0.027	<0.001	Academic performance	−0.142	<0.001
SDQ—Peer relationship problems					
Model 1	0.010	<0.001	Time spent (weekdays)	0.087	0.001
			Academic performance	−0.056	0.008
SDQ—Hyperactivity/inattention					
Model 1	0.021	<0.001	Physical activity level	0.127	<0.001
			Academic performance	−0.089	<0.001
SDQ—Prosocial behavior					
Model 1	0.049	<0.001	Academic performance	0.103	<0.001
			Physical activity level	0.065	0.002
			Time spent (weekend)	−0.074	0.006
			Money spent (month)	−0.089	<0.001

## Data Availability

The database can be accessed through the corresponding author.

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
