# Peer review of "Influence of Gender, Parental Control, Academic Performance and Physical Activity Level on the Characteristics of Video Game Use and Associated Psychosocial Problems in Adolescents"

_behavsci, 2024, doi:10.3390/bs14121204_

Round 1
Reviewer 1 Report
Comments and Suggestions for Authors
This is an interesting study investigating important factors predicting. adolescents' gaming behaviors and associated psychosocial issues. The sample size is robust and the methods are appropriate. More importantly, the topic is timely. I enjoyed reading the paper and agree that it may contribute well to the literature. I have several comments to improve the manuscript further:
1. It will be good to enhance the theoretical foundation of the manuscript. While there are references to the impact of gaming on psychological health, there’s no integration of relevant psychological theories (e.g., Self-Determination Theory, Uses and Gratifications Theory) to frame why adolescents are drawn to different types of games or how these choices relate to their psychosocial outcomes. Incorporating these theories would enhance the depth of the discussion and provide a stronger foundation for the research questions.
The motivational pull of video games: A self-determination theory approach. Motivation and emotion, 30, 344-360.
2. The analysis shows that males play more video games, but the discussion is limited. It will be useful to expand the discussion on why these gender differences exist. Are the differences purely based on social norms and stereotypes, or are there other factors at play? The paper should explore the motivational and contextual reasons behind these gender differences, using literature that discusses how gaming is marketed and perceived differently across genders.
3. One noteworthy limitation of this study is the lack of consideration for gaming motivations. While the study explores gender, parental control, academic performance, and physical activity, it overlooks the reasons adolescents engage in gaming, which can significantly influence whether gaming behaviors are adaptive or problematic. I recommend that the authors address this limitation in the discussion. Nevertheless, the examination of contextual factors such as type of games, the social aspects, and the timing is commendable and important. A review paper that might be relevant for the discussion: A critical review on the moderating role of contextual factors in the associations between video gaming and well-being. Computers in Human Behavior Reports, 4, 100135.
4. The hypotheses presented are somewhat vague, particularly regarding academic performance and physical activity. It would be beneficial to state these more clearly and ensure they are directly testable.
5. Given the cross-sectional nature of the study, I would like the authors to acknowledge the possibility of reverse causation as a possible explanation of the results
Author Response
Reviewer 1
This is an interesting study investigating important factors predicting. adolescents' gaming behaviors and associated psychosocial issues. The sample size is robust and the methods are appropriate. More importantly, the topic is timely. I enjoyed reading the paper and agree that it may contribute well to the literature. I have several comments to improve the manuscript further:
+ Dear reviewer. Thank you very much for your time and your work reviewing out manuscript. We will try to answer all the questions you indicate in order to substantially improve the manuscript.
- It will be good to enhance the theoretical foundation of the manuscript. While there are references to the impact of gaming on psychological health, there’s no integration of relevant psychological theories (e.g., Self-Determination Theory, Uses and Gratifications Theory) to frame why adolescents are drawn to different types of games or how these choices relate to their psychosocial outcomes. Incorporating these theories would enhance the depth of the discussion and provide a stronger foundation for the research questions.
The motivational pull of video games: A self-determination theory approach. Motivation and emotion, 30, 344-360.
+ Thank you very much for your contribution. The introduction has been complemented with these psychological theories to give more theoretical support to the research.
- The analysis shows that males play more video games, but the discussion is limited. It will be useful to expand the discussion on why these gender differences exist. Are the differences purely based on social norms and stereotypes, or are there other factors at play? The paper should explore the motivational and contextual reasons behind these gender differences, using literature that discusses how gaming is marketed and perceived differently across genders.
+ Thank you very much for your contribution. This information has been included in the discussion, indicating the possible reasons for these differences (marketing, motivations, barriers, or stereotypes).
- One noteworthy limitation of this study is the lack of consideration for gaming motivations. While the study explores gender, parental control, academic performance, and physical activity, it overlooks the reasons adolescents engage in gaming, which can significantly influence whether gaming behaviors are adaptive or problematic. I recommend that the authors address this limitation in the discussion.
+ Thank you very much for your great contribution. We have included these aspects in the limitations, as it is an important issue to be considered in future papers.
Nevertheless, the examination of contextual factors such as type of games, the social aspects, and the timing is commendable and important. A review paper that might be relevant for the discussion: A critical review on the moderating role of contextual factors in the associations between video gaming and well-being. Computers in Human Behavior Reports, 4, 100135.
+ Thank you very much. We have included the most relevant information of this interesting study in different sections of the discussion to complement it in greater detail.
- The hypotheses presented are somewhat vague, particularly regarding academic performance and physical activity. It would be beneficial to state these more clearly and ensure they are directly testable.
+ Thank you very much for your contribution. We have tried to rewrite the hypotheses to be more specific in relation to the research aims.
- Given the cross-sectional nature of the study, I would like the authors to acknowledge the possibility of reverse causation as a possible explanation of the results.
+ Thank you very much for your comment. This has been included in the ‘Limitations’ section.
+ Dear reviewer, thank you very much for your work and contributions. We have carried out all your indications, but we remain at your disposal for any other aspect that needs to be modified.
Reviewer 2 Report
Comments and Suggestions for Authors
This is a primarily descriptive research article that involves a multitude of variables. The authors have written in a clear and standardized manner. However, upon overall consideration, the significance of the article seems quite limited. I offer some suggestions in hopes that they will be helpful for the authors' future submissions.
1. The authors included a large number of variables in their study, including gender, environment, academics, and physical activities, but the most critical point is not explained—why these variables were chosen? The statement of previous research does not illustrate the importance of these variables. The presence of numerous variables without a solid rationale makes the study appear disjointed and fails to highlight the significance of the research.
2. Formulating hypotheses requires theoretical basis and reasoning. For this study, I believe the authors could just state the questions they intend to explore in the introduction.
3. Rstudio does not seem to be a statistical software. The authors need to specify what tools (such as a particular package) were used to calculate the sample size.
4. The results section is very rich in content, and the authors have presented the results in great detail. However, in the discussion section, this straightforward presentation leads to a lack of clarity in logic. Please use at least second-level or third-level headings to make the structure of the discussion section more apparent.
5. Although the study has yielded rich results, the contribution of the research is still not clear. It is recommended to use a separate subsection in the discussion to describe the research implication.
6. I also suggest that, the authors could re-think their research question and construct and test potential models to further examine the relations among the variables.
Author Response
Reviewer 2
This is a primarily descriptive research article that involves a multitude of variables. The authors have written in a clear and standardized manner. However, upon overall consideration, the significance of the article seems quite limited. I offer some suggestions in hopes that they will be helpful for the authors' future submissions.
+ Dear reviewer, thank you very much for your work and for helping us to improve the manuscript. We will take care of everything you tell us to improve it considerably.
- The authors included a large number of variables in their study, including gender, environment, academics, and physical activities, but the most critical point is not explained—why these variables were chosen? The statement of previous research does not illustrate the importance of these variables.The presence of numerous variables without a solid rationale makes the study appear disjointed and fails to highlight the significance of the research.
+ Thank you very much for your comment. We have provided in the introduction the relevance of each of the variables in the study in order to have a more consistent theoretical basis for our study.
- Formulating hypotheses requires theoretical basis and reasoning. For this study, I believe the authors could just state the questions they intend to explore in the introduction.
+ Dear reviewer, agreeing with your comment, given that the other two reviewers have asked to expand and split the hypotheses, we have chosen to expand and split the hypotheses where there was a good theoretical basis, and to leave without hypotheses those questions where there is insufficient basis. We hope that this intermediate option will satisfy all reviewers.
- Rstudio does not seem to be a statistical software. The authors need to specify what tools (such as a particular package) were used to calculate the sample size.
+ Thank you very much for your contribution. We have included the corresponding statistical package used, as well as the references on which we have based our sample size calculation with the R software.
- The results section is very rich in content, and the authors have presented the results in great detail. However, in the discussion section, this straightforward presentation leads to a lack of clarity in logic. Please use at least second-level or third-level headings to make the structure of the discussion section more apparent.
+ Thank you very much for your great contribution. Second and third level sub-sections have been included throughout the discussion to make it easier for the reader to understand and follow.
- Although the study has yielded rich results, the contribution of the research is still not clear. It is recommended to use a separate subsection in the discussion to describe the research implication.
+ Thank you very much for your indication. A specific section has been included in the discussion for ‘theoretical and practical implications’.
- I also suggest that the authors could re-think their research question and construct and test potential models to further examine the relations among the variables.
+ Thank you very much for your contribution. We have carried out correlation analysis, as well as linear regression, between the quantitative variables to find out the association between them. This has been included in the introduction, as an aim, as well as in the results and discussion.
+ Dear reviewer, thank you for reviewing our manuscript to help us improve it. We have responded to all your suggestions. We remain at your disposal should you consider it necessary to make any further modifications.
Reviewer 3 Report
Comments and Suggestions for Authors
Thank you very much for the opportunity to review an exceptionally interesting article on the demographic and personality characteristics of adolescents who play video games.
The article is excellently written and highly in-depth. It includes an impressive sample of 2,567 adolescents from Spain, which allows for a considerable generalization of the study's findings.
All sections are clear and understandable. The statistical methods used are appropriate and well-justified. The discussion is thorough and nicely summarizes and interprets all obtained results.
I believe the article is suitable for publication in its current form. However, I would suggest that the authors consider dividing some of the hypotheses so they don't encompass so many relationships at once (e.g., hypothesis 2 could be split into several hypotheses based on the criterion variable included—so that there would be a separate hypothesis for each criterion).
Author Response
Reviewer 3
Thank you very much for the opportunity to review an exceptionally interesting article on the demographic and personality characteristics of adolescents who play video games.
The article is excellently written and highly in-depth. It includes an impressive sample of 2,567 adolescents from Spain, which allows for a considerable generalization of the study's findings.
+ Thank you very much.
All sections are clear and understandable. The statistical methods used are appropriate and well-justified. The discussion is thorough and nicely summarizes and interprets all obtained results.
+ Thank you.
I believe the article is suitable for publication in its current form. However, I would suggest that the authors consider dividing some of the hypotheses so they don't encompass so many relationships at once (e.g., hypothesis 2 could be split into several hypotheses based on the criterion variable included—so that there would be a separate hypothesis for each criterion).
+ Dear reviewer, thank you very much for your contribution. Each of the hypotheses has been divided into parts for better understanding and follow-up.
Round 2
Reviewer 1 Report
Comments and Suggestions for Authors
The authors have addressed all my comments well. The paper is ready for publication. Congratulations!
Author Response
Thank you very much for your help and input to improve the manuscript.
Reviewer 2 Report
Comments and Suggestions for Authors
Thank you for your careful revision and response. however, I still have some suggestions for you:
(1) the contents and research questons are too many, it seems that this manuscript is data-drived without clear research question; at the same time, you examine the general use and addiction in this study, it seems confused and uncalear, maybe you could focus on the addictive and problematic use, or you could focus on some of the current independent variables;
(2) the subtitles are strongly needed in the intrduction;
(3) the research question should be more focuse based on previous studies and research gap;
(4) regading the data analysis and the findings, these contents are lengthy, they should be focused on the main research question and represent the mian findings, for example not all the differences shoudl be analyzed and reported;
(5) the cotents and language could be streamlined and more clear;
(6) in summary, I suggest that you could highlight and clearly state the mian researh question, and all the contents should be directly related to the main resarch question.
Author Response
Thank you for your careful revision and response. however, I still have some suggestions for you:
(1) the contents and research questions are too many, it seems that this manuscript is data-derived without clear research question; at the same time, you examine the general use and addiction in this study, it seems confused and unclear, maybe you could focus on the addictive and problematic use, or you could focus on some of the current independent variables;
+ Thank you very much for your great contribution. We have rewritten the introduction so that it is more focused on the variables discussed in the article. It is true that before it was more diffuse and did not properly address the issue at hand. We have also eliminated the hypotheses, replacing them with research questions. It has also been specified that previous scientific literature is based on problematic and addictive use of video games, and that our approach is more open and seeks to understand differences in other psychological variables related to video games.
(2) the subtitles are strongly needed in the introduction;
+ Thank you very much. Subtitles have been included in the introduction to better guide the reader.
(3) the research question should be more focus based on previous studies and research gap;
+ Thank you very much. The research question has been defined more rigorously and on the basis of what is known from previous research.
(4) regarding the data analysis and the findings, these contents are lengthy, they should be focused on the main research question and represent the main findings, for example not all the differences should be analyzed and reported;
+ Thank you very much for this contribution. We have considerably reduced the paragraphs in the results section, as well as the discussion section, by deleting those paragraphs that were not so relevant to the research questions.
(5) the contents and language could be streamlined and more clear;
+ Thank you very much. The manuscript has been sent to an expert translator of scientific articles for English revision.
(6) in summary, I suggest that you could highlight and clearly state the main research question, and all the contents should be directly related to the main research question.
+ Thank you very much for your input. We have tried to be more concise in the manuscript and to stick to the research question with our statements. We remain at your disposal for any further modifications you may consider necessary.
Round 3
Reviewer 2 Report
Comments and Suggestions for Authors
I appreciate it very much for your careful revisions and responses. I only have two minor suggestions:
(1) in the abstract, the aim of the study should be briefly stated, and the significance or implications could also be briefly stated;
(2) the causal statements throughout the manuscript should be avoided due to the cross-sectional design
Author Response
I appreciate it very much for your careful revisions and responses. I only have two minor suggestions:
(1) in the abstract, the aim of the study should be briefly stated, and the significance or implications could also be briefly stated;
- Thank you very much. This information has been included in the abstract.
(2) the causal statements throughout the manuscript should be avoided due to the cross-sectional design.
- Thank you very much for your contribution. The entire manuscript has been revised, mainly the results and discussion to correct this.